# Inactivation of a CRF-dependent amygdalofugal pathway reverses addiction-like behaviors in alcohol-dependent rats

Giordano de Guglielmo [1], Marsida Kallupi [1], Matthew B. Pomrenze[2], Elena Crawford[1], Sierra Simpson [1], Paul Schweitzer[1], George F. Koob[3], Robert O. Messing [2] & Olivier George [1]

The activation of a neuronal ensemble in the central nucleus of the amygdala (CeA) during alcohol withdrawal has been hypothesized to induce high levels of alcohol drinking in dependent rats. In the present study we describe that the CeA neuronal ensemble that is activated by withdrawal from chronic alcohol exposure contains ~80% corticotropin-releasing factor (CRF) neurons and that the optogenetic inactivation of these CeA CRF+ neurons prevents recruitment of the neuronal ensemble, decreases the escalation of alcohol drinking, and decreases the intensity of somatic signs of withdrawal. Optogenetic dissection of the downstream neuronal pathways demonstrates that the reversal of addiction-like behaviors is observed after the inhibition of CeA CRF projections to the bed nucleus of the stria terminalis (BNST) and that inhibition of the CRF$^{CeA-BNST}$ pathway is mediated by inhibition of the CRF-CRF$_1$ system and inhibition of BNST cell firing. These results suggest that the CRF$^{CeA-BNST}$ pathway could be targeted for the treatment of excessive drinking in alcohol use disorder.

[1] Department of Neuroscience, The Scripps Research Institute, La Jolla, CA 92037, USA. [2] Departments of Neuroscience and Neurology and the Waggoner Center for Alcohol and Addiction Research, The University of Texas at Austin, Austin, TX, USA. [3] National Institute on Drug Abuse, National Institutes of Health, Baltimore, MD 21224, USA. Correspondence and requests for materials should be addressed to O.G. (email: ogeorge@scripps.edu)

Alcohol addiction is a chronic relapsing disorder that is associated with compulsive drinking, the loss of control over intake, and the emergence of a negative emotional state during abstinence from alcohol[1]. Animal and human studies suggest a key role for the central nucleus of the amygdala (CeA) in alcohol use disorders[2–4]. Chronic alcohol use alters CeA neuronal transmission[5–9], and the CeA has been shown to encode alcohol-related memories[10]. The activation of a specific neuronal ensemble in the CeA during alcohol withdrawal is associated with high levels of alcohol drinking in alcohol-dependent rats[4]. However, very little is known about the cellular phenotypes of these neurons and the brain regions that are controlled by this CeA neuronal ensemble.

Converging lines of evidence suggest that both alcohol and alcohol withdrawal affect CRF neurotransmission in the CeA[11,12]. Extracellular levels of CRF in the CeA increase during withdrawal from chronic alcohol exposure[1,13–15], and systemic or intra-CeA administration of specific $CRF_1$ receptor antagonists reduces negative emotional states that are associated with alcohol withdrawal and excessive alcohol drinking in dependent rats[15–18]. However, no direct evidence has been reported that CeA CRF neurons are responsible for these behaviors. Indeed, the activation of CeA $CRF_1$ receptors could result from the activation of CRF neurons that are located in other brain regions that project to the CeA, such as the bed nucleus of the stria terminalis (BNST), lateral hypothalamus (LH), and parasubthalamic nucleus (pSTN).

We hypothesized that CeA CRF neurons constitute a major portion of the CeA neuronal ensemble that is recruited during alcohol withdrawal. To test this hypothesis, we used Crh-Cre transgenic rats[19] that were tested in an animal model of alcohol dependence combined with in vivo optogenetics and immediate early gene brain mapping. We then dissected the role of the different downstream CeA-CRF pathways in alcohol dependence using the optogenetic inactivation of terminals from CeA-CRF projections[19]. The results show that the withdrawal-activated neuronal ensemble in the CeA contains largely CRF neurons (~80%) and that inactivation of these CeA CRF+ neurons prevents recruitment of the CeA neuronal ensemble, decreases dependence-induced alcohol drinking, and reduces the intensity of alcohol withdrawal signs. Moreover, the optogenetic dissection of downstream neuronal pathways demonstrates that this effect requires the activation of CeA CRF+ terminals in the BNST.

## Results

### CeA CRF neurons are recruited during alcohol withdrawal.
We first characterized the cellular phenotype of CeA neurons that are activated during alcohol withdrawal[4]. Rats were trained to self-administer alcohol and then were made alcohol-dependent by exposure to chronic intermittent alcohol vapor to induce the escalation of alcohol intake. After the stabilization of escalated drinking (Fig. 1a), the animals were transcardially perfused with 4% buffered paraformaldehyde. The brains were harvested, postfixed, and sectioned at 40 μm. Withdrawal from alcohol vapor significantly increased the number of Fos+ neurons in the CeA compared with naive rats (Fig. 1b). This increase was limited to a small subpopulation of neurons in the CeA. The lateral CeA (which contains the majority of CRF neurons[19]) exhibited the largest increase in Fos+ neurons (Supplementary Figure 1). Double-labeling with Fos/NeuN revealed that only 7–8% of all CeA neurons (NeuN+) were also Fos+ (Supplementary Figure 1). Double Fos/CRF immunohistochemistry showed that the total number of CRF neurons did not change between alcohol-withdrawn and alcohol-naive rats (Fig. 1c), but the number of Fos+/CRF+ neurons in the CeA significantly increased during

withdrawal (Fig. 1d) and accounted for most of the Fos+ neurons (78.5 ± 3.8%).

### Halorhodopsin inhibits CeA CRF neurons.
To control the activity of CeA CRF neurons, we selectively targeted them by injecting AAV5-EF1a-DIO-NpHR-eYFP in the CeA in Crh-Cre rats[19]. The rats exhibited the Cre-dependent expression of enhanced yellow fluorescent protein (eYFP) after the infusion of AAV5-EF1a-DIO-NpHR-eYFP in the CeA (Fig. 2a). Confocal analysis at 63× magnification, followed by the three-dimensional reconstruction of neuronal cell bodies and branches (Fig. 2b), revealed that 99% of eYFP+ neurons were CRF-immunoreactive in the cell soma or branches[19]. To validate our approach and confirm the efficacy of NpHR in preventing CRF+ neuron firing, electrophysiological recordings of neuronal activity from single neurons were performed in acute CeA slices from Crh-Cre rats. Central nucleus of the amygdala neurons were depolarized by a current injection to evoke the sustained firing of action potentials. CRF+ neurons were identified by YFP fluorescence[20]. In six neurons that showed fluorescence, the resting membrane potential was −64.3 ± 1.8 mV, and the input resistance was 276 ± 20 MΩ. Exposure to light at 15 Hz from a green laser (532 nm)[16] for 6.5 s elicited an average hyperpolarization of 6.5 ± 1.0 mV (from −52.9 ± 0.4 to −59.4 ± 1.0 mV), concomitant with a decrease in action potential firing from 2.4 ± 0.35 to 0.25 ± 0.13 Hz (Fig. 2c). In another six neurons that did not show fluorescence, the resting membrane potential was −65.6 ± 1.7 mV, and the input resistance was 264 ± 15 MΩ, similar to the values that were obtained in fluorescent neurons. However, exposure to the laser (6.5 s at 15 Hz) did not significantly alter the membrane potential (51.8 ± 0.5 mV vs. 52.3 ± 0.3 mV) or the firing rate (2.02 ± 0.28 Hz vs. 1.98 ± 0.36 Hz; Fig. 2d). This experiment was repeated with sustained 5 min illumination of green light (HaloR, 532 nm) to mimic the in vivo behavioral paradigm of light exposure. Exposure to 5 min continuous green light (HaloR, 532 nm)[16] elicited an average hyperpolarization of 8.5 ± 1.3 mV (from −52.5 ± 0.4 to −61 ± 1.2 mV), concomitant with a robust decrease in action potential firing from 1.03 ± 0.13 to 0.4 ± 0.07 Hz (Supplementary Figure 4A, B).

### Optogenetic inhibition of CeA CRF neurons.
We next investigated the effect of the bilateral optogenetic inhibition of CeA CRF neurons on alcohol drinking in dependent and nondependent rats. Crh-Cre rats were bilaterally infused in the CeA with AAV5-EF1a-DIO-NpHR-eYFP or AAV5-EF1a-DIO-eYFP and implanted with bilateral optical fibers in the CeA. After the acquisition of alcohol self-administration, the optogenetic inhibition of CeA CRF neurons was induced during the alcohol self-administration session. In nondependent animals, CeA CRF neuron inhibition (laser ON) did not affect drinking in control rats that expressed eYFP or in NpHR-expressing rats. Water self-administration was also unaffected by the green laser (Fig. 3b). After 6 weeks of chronic intermittent alcohol vapor exposure, the animals escalated their alcohol consumption (Fig. 3c). Activation of the green laser selectively reduced alcohol self-administration in NpHR-injected rats, without affecting water self-administration (Fig. 3d).

To verify that the effect of optical inhibition was not attributable to nonspecific effects of prolonged (30 min) illumination of the CeA with the green laser, we measured alcohol drinking after only 5 min of illumination. There was no difference between the magnitude of the effect during the first 5 min and the entire 30-min session (Supplementary Figure 2A). Activation of the green laser did not affect saccharin intake in control or NpHR-expressing rats (Fig. 3e), indicating that the effect of CeA CRF neuronal inhibition was specific to escalated alcohol intake. Activation of the green laser also did not affect the total distance

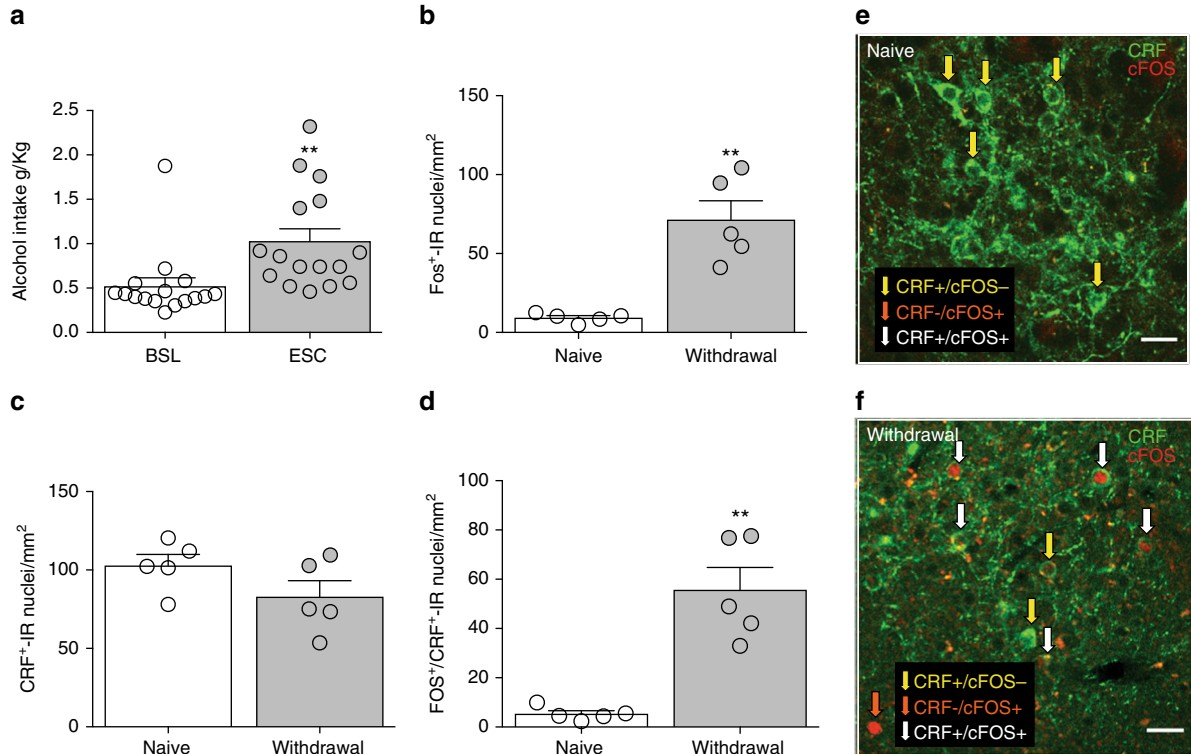

**Fig. 1** Alcohol withdrawal activated CRF neurons in the CeA. **a** Exposure to chronic intermittent alcohol vapor increased responding for alcohol 2.5-fold compared with the level of baseline responding before vapor exposure ($t_8 = 4.38$, **$p < 0.01$, escalation [ESC] vs. baseline [BSL], paired $t$-test). **b** Withdrawal from alcohol vapor significantly increased the number of Fos+ neurons in the CeA ($t_8 = 5.12$, **$p < 0.001$, unpaired $t$-test). **c** Double Fos-CRF immunohistochemistry showed that the total number of CRF neurons did not significantly change between alcohol-withdrawn and naive rats ($t_8 = 1.14$, $p > 0.05$, unpaired $t$-test), but **d** the number of Fos+/CRF+ neurons in the CeA significantly increased during withdrawal ($t_8 = 5.45$, $p < 0.001$, unpaired $t$-test) and represented the majority of Fos+ neurons (80%). **e, f** Representative images of double Fos (red)-CRF (green) immunohistochemistry in the CeA in alcohol-naive (**e**) and alcohol-withdrawn (**f**) rats. Yellow arrows indicate Fos-/CRF+ cells. White arrows indicate Fos+/CRF+ cells. Orange arrows indicate Fos+/CRF− cells. The data are expressed as mean ± SEM. Scale bars = 50 μm

traveled (Fig. 3f) or time spent in the center (Fig. 3g) of the open field in control rats or NpHR-expressing rats, indicating that the effects of laser stimulation on alcohol drinking were not attributable to nonspecific locomotor effects.

**Effects of unilateral inhibition of CeA CRF neurons.** A group of *Crh*-Cre rats ($n = 12$) was bilaterally infused with AAV5-EF1a-DIO-NpHR-eYFP and implanted with unilateral optical fibers in the CeA. Illumination with the green laser significantly reduced alcohol self-administration (Fig. 4c). This effect was reversible across 6 days of alternating the sessions with the laser ON and OFF. Immediately before the last two alcohol self-administration sessions, the rats were observed for somatic signs of withdrawal. As shown in Fig. 4d, the laser ON group exhibited significant decreases in ventromedial limb retraction and abnormal gait. The sum of the five rating scores revealed a significant decrease in overall withdrawal severity (Fig. 4d, inset) during the unilateral inhibition of CeA-CRF neurons. Saccharin self-administration was unaffected by the unilateral inhibition of CeA CRF neurons (Fig. 4e). A separate group of rats ($n = 15$) was bilaterally infused with AAV5-EF1a-DIO-NpHR-eYFP and implanted with unilateral optical fibers in the CeA to test the role of CeA CRF neurons in the recruitment of the CeA neuronal ensemble during withdrawal from chronic intermittent alcohol exposure. Rats were made dependent using the same chronic intermittent alcohol exposure protocol as in the previous experiments. Brains were perfused after 8 h of alcohol withdrawal, 90 min after 30-min exposure to the green laser. The optical inhibition (laser ON) of

CeA CRF neurons prevented the increase in Fos+ neurons that was normally observed during withdrawal (laser OFF). Unilateral inhibition also decreased the number of Fos+ neurons on the contralateral side (Fig. 4f, j).

**The CRF$^{CeA-BNST}$ pathway mediates alcohol drinking.** *Crh*-Cre rats ($n = 12$) were bilaterally infused with AAV5-EF1a-DIO-NpHR-eYFP in the CeA and unilaterally implanted with an optical fiber in the ventral BNST (Fig. 5b). They were then made dependent on alcohol using the chronic intermittent exposure procedure before testing the effect of optogenetic inhibition on alcohol drinking, alcohol withdrawal signs, and saccharin self-administration. The optogenetic inhibition of CRF$^{CeA-BNST}$ terminals reduced alcohol self-administration to levels that were similar to baseline pre-vapor exposure (Fig. 5c). Immediately before the last two alcohol self-administration sessions, the rats were observed for withdrawal signs 8 h into withdrawal. As shown in Fig. 5d, the laser ON group exhibited significant decreases in abnormal gait and body tremors. The sum of the five rating scores revealed a significant decrease in overall withdrawal severity after the inhibition of CRF$^{CeA-BNST}$ terminals. Saccharin self-administration was unaffected by the optogenetic inhibition of CRF$^{CeA-BNST}$ terminals (Fig. 5e).

To determine whether the optogenetic inhibition of CRF$^{CeA-BNST}$ terminals was sufficient to alter the activity of BNST neurons, we prepared a separate cohort of *Crh*-Cre rats ($n = 8$) that were bilaterally infused with AAV5-EF1a-DIO-NpHR-eYFP in the CeA. We prepared BNST slices and recorded

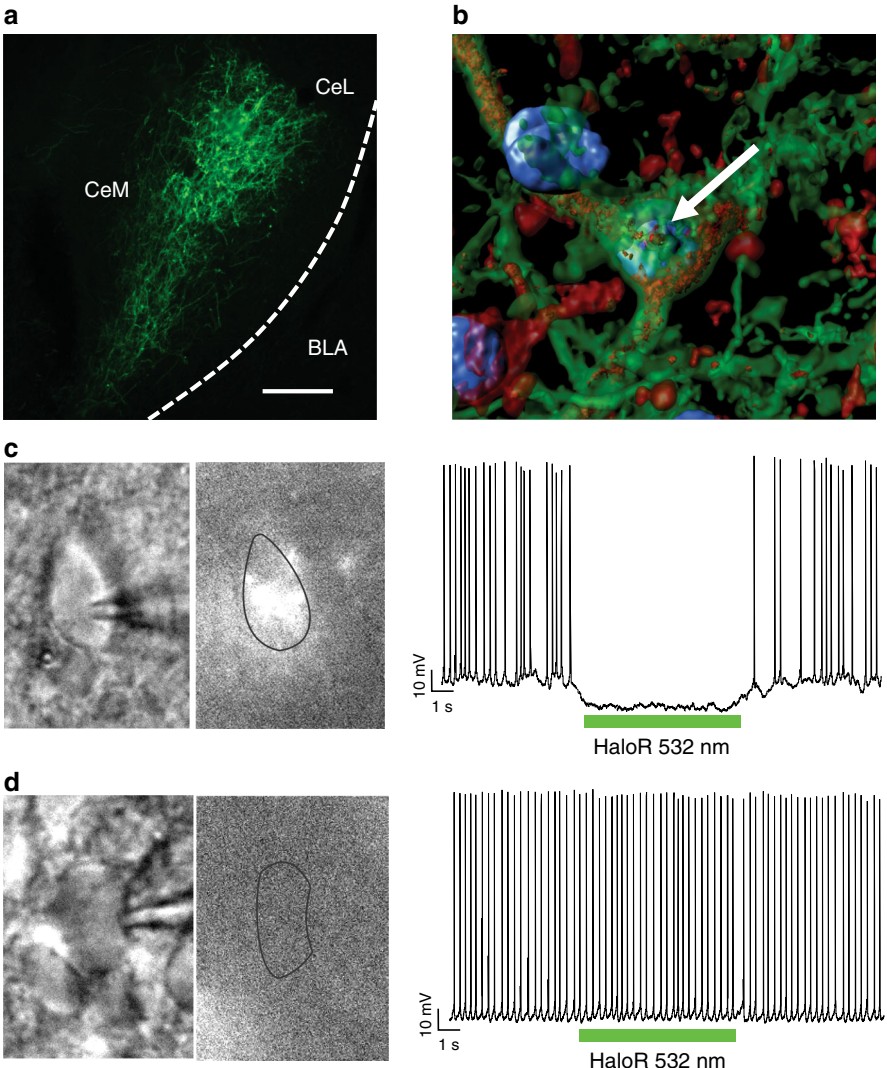

**Fig. 2** Validation of the *Crh*-Cre rat. **a** Cre-dependent eYFP expression in the CeA. Scale bar = 200 μm. **b** Rendered isosurface analysis demonstrated the colocalization of CRF immunoreactivity within CeA neurons that also expressed Cre-dependent eYFP. The arrow points to an example of eYFP and CRF in the same neuron. **c** Infrared (Left) and eYFP fluorescence (Middle) images of a CeA neuron. (Right) Current clamp recording of the same neuron (manually depolarized to −50 mV), showing hyperpolarization and the suppression of firing in response to the delivery of a 6.5 s train (asterisks) of green light (HaloR, 532 nm). **d** Representative images of a non-fluorescent CeA neuron and accompanying recording (manually depolarized to −52 mV) and a recording of that neuron that shows no response to light stimulation

from cells in the BNST. In eight of 14 BNST neurons (resting membrane potential −66.7 ± 1.4 mV), exposure to light from a green laser (HaloR, 532 nm) at 15 Hz for 6.5 s decreased action potential firing from 0.37 ± 0.06 to 0.22 ± 0.05 Hz (Supplementary Figure 8). However, in the other six neurons (resting membrane potential −64.6 ± 2.1 mV), similar laser exposure led to a less than 10% decrease (0.49 ± 0.09 Hz vs. 0.46 ± 0.03 Hz) in the firing rate, and we considered these neurons "non-responders" (Supplementary Figure 8). This experiment was repeated with 5 min of exposure to green light (HaloR, 532 nm) to mimic the behavioral paradigm of light exposure. Continuous 5 min green light exposure (HaloR, 532 nm)[16] elicited a decrease in action potential firing from 0.91 ± 0.14 to 0.6 ± 0.1 Hz (Fig. 5f, g, Supplementary Figure 4C, D).

To determine the influence of CRF itself on the effects of the optogenetic inhibition of CRF$^{CeA-BNST}$ terminals, we prepared a separate cohort of *Crh*-Cre rats ($n = 8$) that were bilaterally infused with AAV5-EF1a-DIO-NpHR-eYFP in the CeA. We prepared BNST slices as in the experiment above, with the difference that optical inhibition was performed in slices that

were pretreated with the CRF$_1$ receptor antagonist R121919. The bath application of R121919 on the BNST slices for 15 min significantly decreased action potential firing from 0.71 ± 0.13 to 0.43 ± 0.11 Hz. Exposure to 5 min continuous green light (HaloR, 532 nm) in the presence of R121919 did not significantly affect the firing rate (from 0.42 ± 0.11 to 0.37 ± 0.10 Hz), suggesting that R121919 occluded the effect of optogenetic inhibition and indicating that the reduction of action potential firing that was observed in BNST neurons after the optogenetic inhibition of CRF$^{CeA-BNST}$ terminals (Fig. 5f, g) was mainly driven by inhibition of the CRF-CRF$_1$ system.

In contrast to the effects in the BNST, the inhibition of CeA-CRF projections to the substantia innominata (CRF$^{CeA-SI}$), lateral hypothalamus/parasubthalamic nucleus (CRF$^{CeA-LH/pSTN}$), and parabrachial nucleus (CRF$^{CeA-PBN}$) did not alter alcohol drinking in dependent rats (see Supplementary Notes, Supplementary Figures 5, 6, 7). As expected, the inhibition of CRF$^{CeA-BNST}$ terminals in rats that were infused with the control virus (AAV5-DIO-eYFP) did not alter alcohol consumption (Fig. 5h).

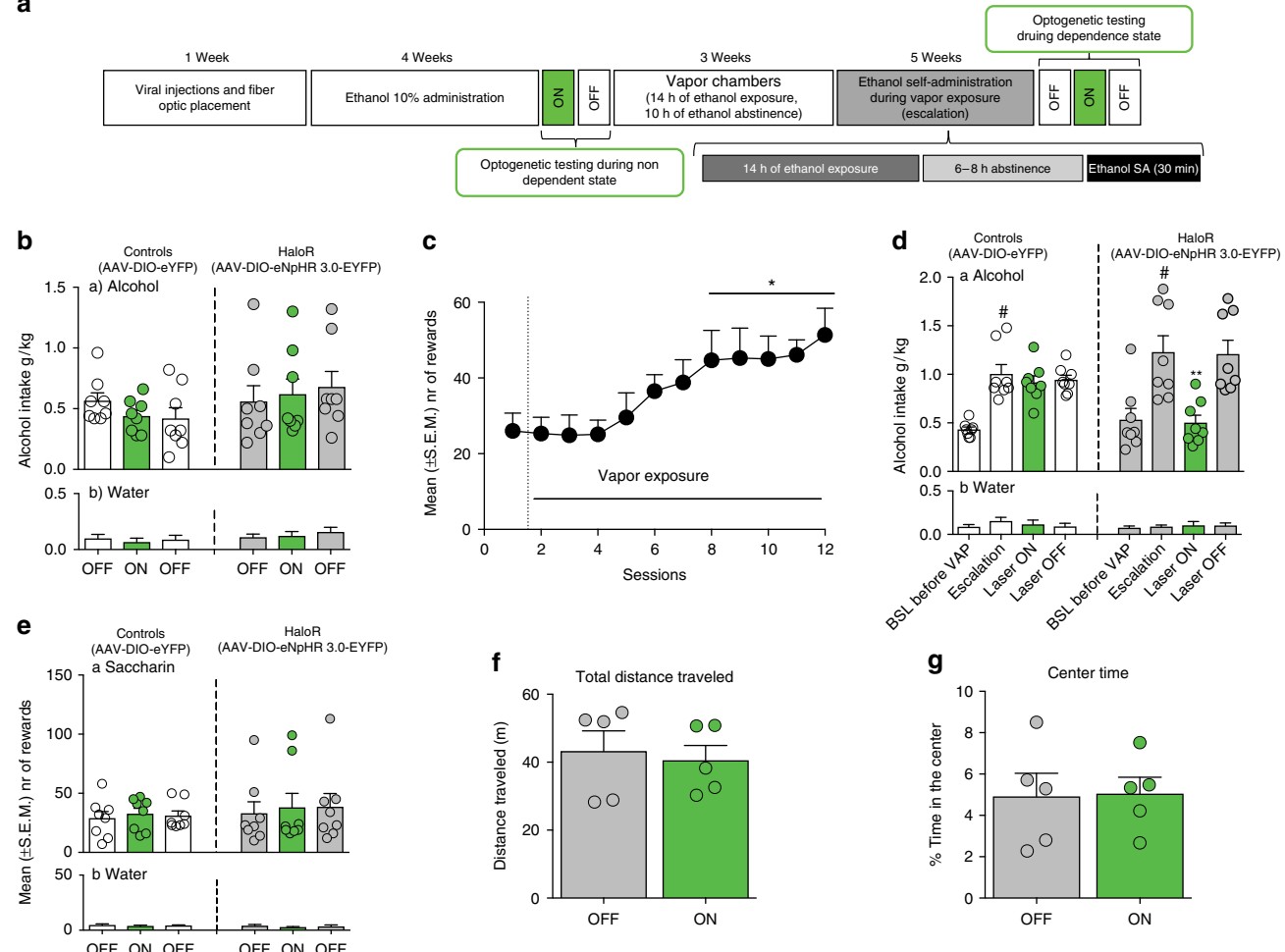

**Fig. 3** Optogenetic inhibition of CeA CRF neurons reduced alcohol intake in alcohol-dependent rats. **a** Timeline of the experiment. **b** Optogenetic inhibition of CeA CRF neurons did not alter alcohol (**a**) or water (**b**) self-administration in nondependent rats. There was no virus × laser treatment interaction ($F_{2,28}$ = 2.10, $p > 0.05$, two-way ANOVA). Water self-administration was also unaffected by the green laser, with no virus × laser treatment interaction ($F_{1,28}$ = 1.07, $p > 0.05$, two-way ANOVA). **c** Alcohol self-administration escalated during intermittent alcohol vapor exposure ($F_{1,28}$ = 0.82, $p < 0.001$, one-way ANOVA), with intake during sessions 8–12 that was significantly greater than during the first day of vapor exposure (*$p < 0.05$, Newman Keuls post hoc test). **d** Effect of optogenetic inhibition of CeA CRF neurons on alcohol (**a**) and water (**b**) self-administration. Both control and NpHR-expressing rats escalated their alcohol intake after 5 weeks of vapor exposure ($F_{1,14}$ = 14.9, $p < 0.01$; $p < 0.05$, vs. baseline pre-vapor, Newman Keuls post hoc test). Activation of the green laser selectively reduced escalated alcohol self-administration only in NpHR-expressing rats. There was a significant virus type × laser treatment interaction ($F_{2,28}$ = 8.38, $p < 0.01$, two-way ANOVA **$p < 0.05$, laser ON vs. OFF, after Newman Keuls) for alcohol, but not for water self-administration ($F_{2,28}$ = 2.44, $p > 0.05$, two-way ANOVA). **e** There were no differences in (**a**) saccharin intake (group × laser treatment interaction: $F_{2,28}$ = 0.06, $p > 0.05$, two-way ANOVA) or (**b**) concomitant water intake (group × laser treatment interaction: $F_{2,28}$ = 0.27, $p > 0.05$, two-way ANOVA). **f**, **g** Optogenetic inhibition of CeA CRF neurons did not alter the (**f**) total distance traveled ($t_8$ = 0.36, $p > 0.05$, paired $t$-test) or (**g**) time spent in the center ($t_8$ = 0.36, $p > 0.05$, paired $t$-test) in the open field in control rats or NpHR-expressing rats. The data are expressed as mean ± SEM

## Discussion

The present study found that CRF neurons comprise the majority of neurons in the CeA neuronal ensemble that is recruited during withdrawal and drives high levels of drinking in alcohol-dependent rats. The optogenetic inhibition of CeA CRF neurons fully reversed the escalation of alcohol drinking and partially alleviated somatic signs of withdrawal in dependent rats, without affecting water or saccharin self-administration. Selective optogenetic inhibition of the CRF$^{CeA-BNST}$ pathway but not CRF$^{CeA-SI}$, CRF$^{CeA-LH/pSTN}$, or CRF$^{CeA-PBN}$ pathway recapitulated the behavioral effects that were observed when CeA CRF neurons were inhibited.

A key aspect of optogenetic approaches is the use of optogenetic and behavioral controls to ensure that the effects are relevant and reproducible. We used several controls to ensure that (i) the AAV5-EF1a-DIO-NpHR-eYFP infusion that was combined

with green laser illumination prevented neuronal firing (Fig. 2), (ii) the AAV5-EF1a-DIO-eYFP control virus that was combined with laser illumination had no effect on its own (Figs. 3d and 5c, Supplementary Figure 5C, 6C, 7C), (iii) the behavioral effects were reversible (Figs. 3d and 5c), (iv) the behavioral effects were observed after both long (30 min) and short (5 min) durations of illumination (Supplementary Figure 2), (v) 5 min of light inhibition resulted in the same effects in slices as 6.5 s, and (vi) the behavioral effects on alcohol drinking in dependent rats were not attributable to nonspecific behavioral effects. Such behavioral controls included water self-administration (Fig. 3), saccharin self-administration (Figs. 3 and 5, Supplementary Figures 5, 6, 7), and locomotor activity (Fig. 3) in alcohol-dependent animals and alcohol self-administration in nondependent animals (Fig. 3). Moreover, the results showed decreases in alcohol drinking in

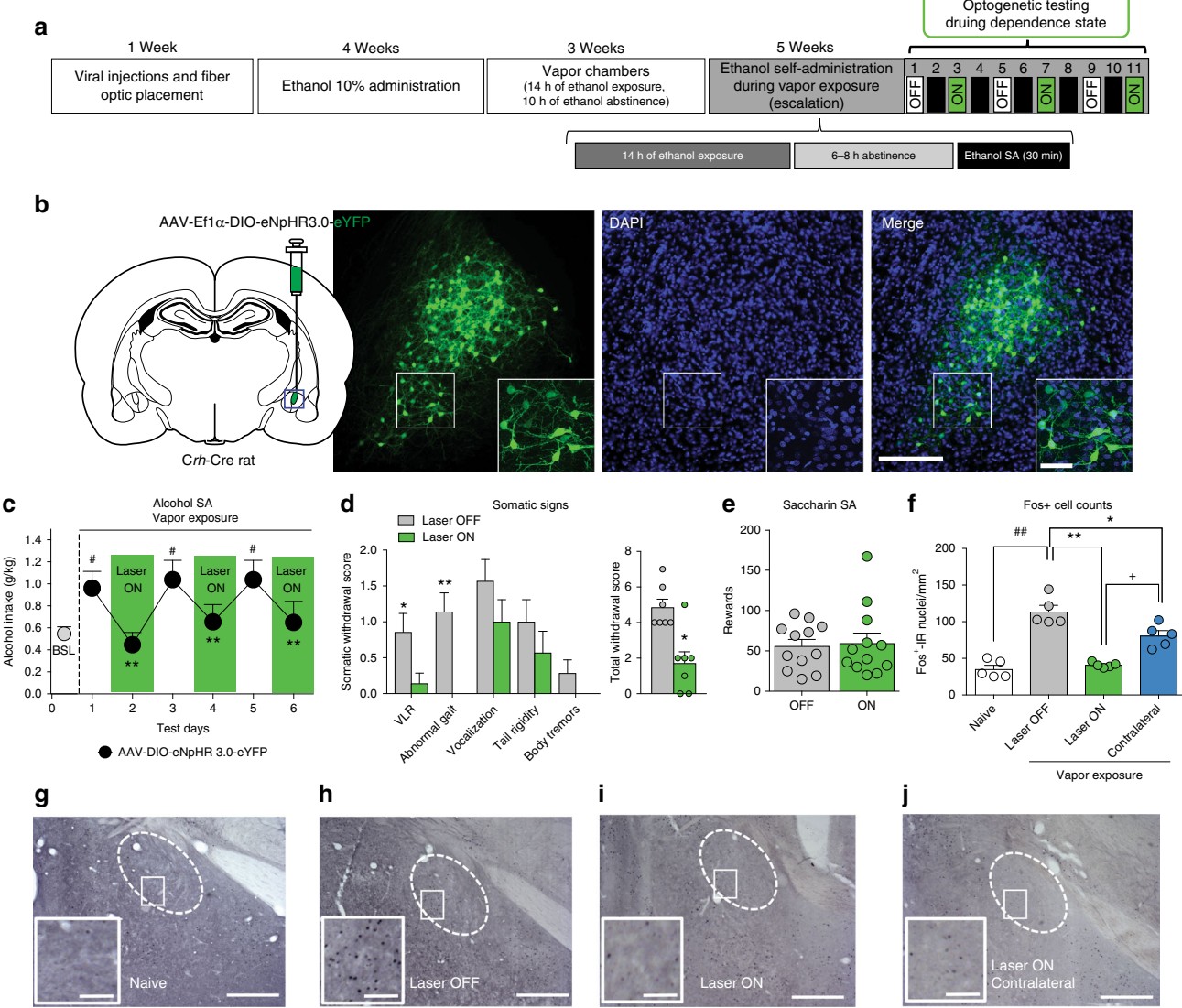

**Fig. 4** Unilateral inhibition of CeA CRF neurons during alcohol withdrawal. **a** Timeline of the experiment. **b** Representative images of NpHR-eYFP expression in the CeA. Scale bars = 200 μm (insets = 50 μm). **c** Time-course of the effect of unilateral optogenetic inactivation of CeA CRF neurons. Rats escalated their alcohol intake after CIE ($F_{6,66} = 9.23$, $p < 0.001$, one-way ANOVA). The escalation of alcohol drinking was observed only on days when the laser was OFF ($p < 0.05$, *vs*. BSL, Newman Keuls post hoc test). A two-way repeated-measures ANOVA revealed a significant time ×laser illumination interaction ($F_{2,22} = 13.15$, $p < 0.001$), with a significant reduction of alcohol self-administration when the green laser was ON compared with when it was OFF ($p < 0.01$, Newman Keuls post hoc test). #$p < 0.05$, *vs*. baseline (BSL); **$p < 0.01$, *vs*. laser OFF. **d** Effect of unilateral CeA CRF neuron inhibition on somatic withdrawal signs. The laser ON group exhibited significant decreases in ventromedial limb retraction (Mann–Whitney $U = 9.00$, $p < 0.05$) and abnormal gait ($U = 3.000$, $p < 0.01$). The sum of the five rating scores revealed a significant decrease in overall withdrawal severity ($U = 4.5$, $p < 0.05$) after the inhibition of CeA-CRF neurons. *$p < 0.05$, *vs*. laser OFF. **e** Unilateral optogenetic inhibition of CeA-CRF neurons did not alter saccharin self-administration ($t_{11} = 0.321$, $p > 0.05$, paired $t$-test). **f** Inhibition of CRF neurons suppressed withdrawal-induced CeA Fos immunoreactivity ($F_{3,16} = 35.27$, $p < 0.001$, one-way ANOVA). The Newman Keuls post hoc test showed that after unilateral inhibition (laser ON), the withdrawal-induced increase in Fos+ neurons (laser OFF; $p < 0.01$, laser OFF vs. naive) was completely prevented in the ipsilateral CeA ($p < 0.01$, laser ON vs. OFF; ##$p < 0.01$, *vs*. naive; **$p < 0.01$, vs. laser OFF) and partially in the contralateral CeA (+$p < 0.05$, ON/contralateral vs. OFF). **g–j** Representative pictures of the different experimental conditions that are depicted in **e**. Scale bars = 200 μm (insets = 50 μm). The data are expressed as mean ± SEM

dependent rats after the bilateral inhibition of CeA CRF neurons, and these effects were replicated by the unilateral inhibition of CeA CRF cell bodies (Fig. 4) and CRF$^{CeA-BNST}$ terminals (Fig. 5).

Previous work has shown that alcohol withdrawal induces Fos expression in CeA neurons[4], but the phenotype of these neurons and the mechanism that is responsible for their activation were unknown. In the present study, we found that such Fos induction is limited to a very small subpopulation (~7–8% of all NeuN+ CeA neurons; Supplementary Figure 1) in all subregions of the CeA, although the central lateral amygdala (CeL; which contains

the majority of CRF neurons[19]) exhibited the largest increase in Fos+ neurons (Supplementary Figure 1). The total number of CRF neurons did not change between groups (Fig. 1), but the number of double-labeled Fos+/CRF+ neurons in the CeA significantly increased during withdrawal from alcohol. Finally, neuronal phenotyping showed that Fos was induced in both CRF+ and CRF− neurons, but CRF+ neurons represented ~80% of the Fos+ population. These findings indicate that most CeA neurons in the ensemble that was activated during alcohol withdrawal expressed CRF. Moreover, the results showed that the

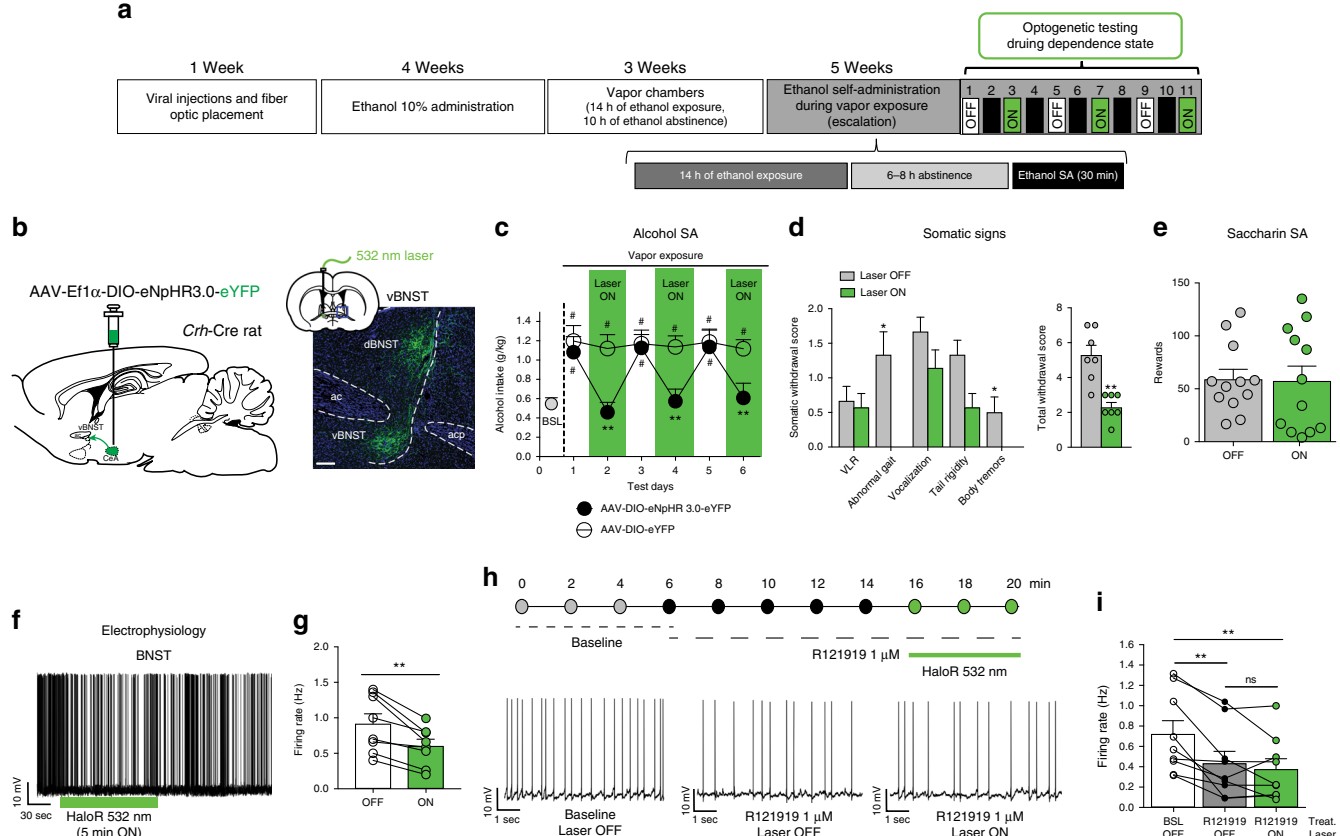

**Fig. 5** Unilateral inhibition of CRF$^{\text{CeA-BNST}}$ terminals during alcohol withdrawal. **a** Timeline of the experiment. **b** Representative images of the areas of injection (CeA) and of optogenetic inhibition (BNST). Scale bar = 200 μm. **c** *Black circles*: effect of optogenetic inactivation of CRF$^{\text{CeA-BNST}}$ terminals in animals injected with AAV-DIO-eNpHR 3.0-eYFP. There was a significant time × laser interaction ($F_{2,22} = 5.31$, $p < 0.05$, two-way ANOVA) with a reduction of alcohol intake when the green laser was turned ON ($p < 0.01$, vs. laser OFF, Newman Keuls post hoc test). Animals escalated their alcohol intake compared with baseline pre-CIE (BSL, $F_{6,66} = 11.43$, $p < 0.0001$, one-way ANOVA) and exhibited escalation only on laser OFF days ($p < 0.001$, Newman Keuls post hoc test) and not after the inhibition of CRF$^{\text{CeA-BNST}}$ terminals. $^{\#\#}p < 0.01$, vs. baseline (BSL); $^{***}p < 0.001$, vs. laser OFF. White circles: laser activation had no effect in AAV-DIO-eYFP-injected rats. Control rats exhibited the escalation of intake ($F_{2,14} = 7.39$, $p < 0.01$, one-way ANOVA) on both laser ON and OFF days (both $p < 0.01$, vs. baseline, Newman Keuls *post hoc* test). $^{\#\#}p < 0.01$, vs. BSL. **d** CRF$^{\text{CeA-BNST}}$ terminal inhibition significantly decreased abnormal gait (Mann–Whitney $U = 6.00$, $p < 0.05$), body tremors ($U = 10.50$, $p < 0.05$), and overall withdrawal severity ($U = 1.5$, $p < 0.01$) (inset). $^*p < 0.05$, $^{**}p < 0.01$, vs. laser OFF. **e** Optogenetic inhibition of CRF$^{\text{CeA-BNST}}$ terminals did not affect saccharin self-administration ($t_{11} = 0.12$, $p > 0.05$, paired $t$-test). **f** Exposure to the green light decreased action potential firing. **g** Inhibition of CRF$^{\text{CeA-BNST}}$ terminals significantly reduced firing frequency ($t_8 = 4.321$, $p < 0.01$, paired $t$-test) in the BNST. **h** (Top) Timeline of the current-clamp recording experiments in the BNST. (Bottom) Representative traces for the experimental conditions. **i** R121919 significantly reduced the firing rate of BNST neurons ($F_{2,16} = 11.63$, $p < 0.001$, one-way ANOVA) and occluded the effect of the optogenetic inhibition of CRF$^{\text{CeA-BNST}}$ terminals ($^{**}p < 0.001$, vs. BSL, Newman Keuls post hoc test). The data are expressed as mean ± SEM

inactivation of CeA CRF neurons completely prevented recruitment of the CeA neuronal ensemble that includes both CRF+ and CRF− neurons. These results are consistent with our previous work that showed that the chemogenetic stimulation of CeA CRF neurons recruited a population of CRF+ and CRF− neurons in the CeL and medial amygdala (CeM) and that the activation of CRF+ and CRF− neurons was prevented by a CRF$_1$ receptor antagonist[19]. Additionally, the unilateral inactivation of CeA CRF neurons was sufficient to partially inhibit Fos expression in the contralateral CeA, indicating that the CeA on both sides is functionally connected and suggesting a possible mechanism to explain the robust effect of unilateral CeA inhibition on alcohol drinking. Central nucleus of the amygdala CRF neurons are γ-aminobutyric acid (GABA)ergic, and the activation of CRF$_1$ receptors in the CeL increases glutamate release locally[11], suggesting that the recruitment of CRF+ and CRF− neurons in the neuronal ensemble may be mediated by the activation of CRF$_1$ receptors on glutamatergic terminals[21]. Moreover, higher

GABAergic transmission in the CeL may also lead to the activation of CeM neurons through disinhibition via GABAergic interneurons[22,23]. A large body of evidence shows that CRF, glutamate, and GABA systems in the CeA play an important role in alcohol dependence[13,15,24,25]. Both acute alcohol withdrawal and activation of the CRF-CRF$_1$ system increase CeA glutamate[8,11] and GABA release in rats[6,7] and mice[26], suggesting that the activation of CRF neurons and recruitment of glutamatergic and GABAergic transmission contribute to recruitment of the CeA neuronal ensemble during alcohol withdrawal.

At the behavioral level, we found that the optogenetic inactivation of CeA CRF neurons reversibly decreased alcohol drinking in dependent rats to levels that were similar to baseline alcohol drinking before the induction of dependence. The inactivation of CeA CRF neurons had no effect on saccharin self-administration or locomotor activity, with no effect of the laser in the control groups. These results suggest that the decrease in drinking that was observed with NpHR was specific to alcohol drinking in

dependent rats and not simply attributable to nonspecific effects on other behaviors. However, still unknown are the downstream neuronal pathways that mediate this behavioral effect.

We recently found that CeA CRF neurons send projections to the BNST, SI, LH/pSTN, and PBN, but the role of each CRF projection in alcohol drinking was unknown. In the present study, we found that inactivation of the CRF[CeA-BNST] projection but not the CRF[CeA-SI], CRF[CeA-LH/pSTN], or CRF[CeA-PBN] projection fully recapitulated the effects of the inactivation of CeA CRF neurons. In slice experiments, we found that the inhibition of CeA CRF axon terminals in the BNST reduced the firing of a subpopulation of BNST neurons. This result is consistent with previous studies that reported that CRF can enhance spontaneous excitatory transmission in the BNST through an increase in glutamate release[27,28]. A similar potentiation of excitatory transmission by CRF has been reported in the CeA and the lateral septum mediolateral nucleus[29]. Importantly, not all of the neurons that were patched in the BNST were responsive to light inhibition. This is consistent with a previous study that showed that the potentiation of excitatory transmission by CRF in the BNST is only observed in a small population of BNST neurons[27]. The reduction of firing in BNST neurons after the inhibition of CeA CRF axon terminals in the BNST was mediated by CRF. Pretreatment with the CRF$_1$ receptor antagonist R121919 mimicked the effect of optogenetic inhibition by reducing action potential firing and occluded the effect of optogenetic inhibition, strongly suggesting that the behavioral effects that were observed after the optogenetic inhibition of CRF CeA neurons and CRF[CeA-BNST] terminals is mediated by the CRF-CRF$_1$ system.

Converging evidence suggests that neuropeptides, including CRF, that are produced locally or project to the BNST are critical for the modulation of neuronal transmission in the BNST and the emergence of fear[30], negative emotional states, and addiction-like behaviors[31–35]. Consistent with these findings, in vivo microdialysis revealed higher extracellular levels of CRF in the BNST and CeA during alcohol withdrawal[13,14]. However, the local administration of a CRF receptor antagonist in the dorsolateral BNST failed to reduce excessive alcohol drinking in dependent rats[16], whereas the administration of a CRF receptor antagonist in the CeA decreased alcohol drinking in dependent rats[9,16]. Nevertheless, the present results demonstrate that CRF terminals in the ventral BNST are critical for excessive alcohol drinking in dependent rats, suggesting that the role of CRF in alcohol drinking may be subregion-specific in the BNST. Alternatively, the effects we observed on alcohol intake may have been partially mediated by other co-neurotransmitters, such as GABA, dynorphin, or somatostatin, considering the high degree of colocalization and overlap between these neurotransmitters in this population of neurons[19,36–38]. Future studies that investigate the Cre-dependent downregulation of GABA, dynorphin, and somatostatin may shed light on this possibility.

The present results also showed that the inhibition of CeA terminals in the BNST completely inhibited the withdrawal-induced escalation of alcohol consumption in dependent rats. The mechanism for this marked reduction of alcohol drinking might be related to a decrease in withdrawal symptoms that leads to less negative reinforcement that is associated with alcohol consumption. However, inhibition had no impact on three of the five withdrawal signs that were evaluated. One explanation for this result may be that somatic signs provide behavioral evidence of alcohol withdrawal[39], but they do not directly measure the negative affective state that is thought to drive the escalation of drinking[40]. Another possibility may be that the CRF[CeA-BNST] pathway plays a role in associative learning, and the inhibition of this pathway inhibits memories of the ameliorative effects of alcohol. Such a mechanism could be involved in mitigating the

escalation of alcohol consumption despite the persistence of some withdrawal signs. Further studies are needed to discriminate between these different possibilities.

In summary, the present study found that the activation of CeA CRF neurons during withdrawal was required for the recruitment of a CeA neuronal ensemble that is responsible for excessive alcohol drinking in dependent rats. Inactivation of the CRF[CeA-BNST] pathway completely reversed the escalation of alcohol drinking during withdrawal and partially prevented the somatic signs of alcohol withdrawal. Inactivation of the CRF[CeA-SI], CRF[CeA-LH/pSTN], and CRF[CeA-PBN] pathways did not impact alcohol drinking or somatic signs of withdrawal. Future studies are needed to (i) examine the role of the CRF[CeA-BNST] pathway in females, (ii) determine whether our negative findings with the CRF[CeA-SI], CRF[CeA-LH/pSTN], and CRF[CeA-PBN] pathways are attributable to technical limitations, (iii) determine whether these three pathways are involved in other aspects of alcohol dependence that were not examined in the present study, and (iv) determine the effect of the optogenetic inhibition of CRF[CeA-BNST], CRF[CeA-SI], CRF[CeA-LH/pSTN], and CRF[CeA-PBN] terminals in nondependent animals.

Overall, the present study identified a neuronal ensemble of alcohol abstinence in the CeA that is under the control of CeA CRF neurons and that activation of the CRF[CeA-BNST] pathway is required for the escalation of alcohol drinking and expression of alcohol withdrawal signs in dependent animals. Finally, we demonstrate that inhibition of the CRF[CeA-BNST] pathway is mediated via inhibition of the CRF-CRF$_1$ system and the inhibition of BNST cell firing. These results identify a critical neuronal ensemble for addiction-like behavior and suggest that the CRF[CeA-BNST] pathway could be targeted for the treatment of excessive drinking in alcohol use disorder.

## Methods

**Subjects**. Adult male *Crh*-Cre rats[19], 2-months-old and weighing 200–225 g at the beginning of the experiments, were housed in groups of two per cage (self-administration groups) in a temperature-controlled (22 °C) vivarium on a 12 h/12 h light/dark cycle (lights on at 10:00 PM) with ad libitum access to food and water. All of the behavioral tests were conducted during the dark phase of the light/dark cycle. All of the procedures adhered to the National Institutes of Health Guide for the Care and Use of Laboratory Animals and were approved by the Institutional Animal Care and Use Committee of The Scripps Research Institute.

**Operant self-administration**. Self-administration sessions were conducted in standard operant conditioning chambers (Med Associates, St. Albans, VT, USA). For the alcohol self-administration studies, the animals were first trained to self-administer 10% (v/v) alcohol and water solutions until a stable response pattern (20 ± 5 rewards) was maintained. The rats were subjected to an overnight session in the operant chambers with access to one lever (right lever) that delivered water (fixed-ratio 1 [FR1]). Food was available ad libitum during this training. After 1 day off, the rats were subjected to a 2 h session (FR1) for 1 day and a 1 h session (FR1) the next day, with one lever delivering alcohol (right lever). All of the subsequent alcohol self-administration sessions lasted 30 min. The rats were allowed to self-administer a 10% (v/v) alcohol solution (right lever) and water (left lever) on an FR1 schedule of reinforcement (i.e., each operant response was reinforced with 0.1 ml of the solution). For the saccharin self-administration study, the rats underwent daily 30 min FR1 sessions. Responses on the right lever resulted in the delivery of 0.1 ml of saccharin (0.04%, w/v). Lever presses on the left lever delivered 0.1 ml of water. This procedure lasted 13 days until a stable baseline of intake was reached.

**Alcohol vapor chambers**. The rats were previously trained to self-administer alcohol in the operant chambers. Once a stable baseline of alcohol intake was reached, the rats were made dependent by chronic intermittent exposure to alcohol vapors. They underwent cycles of 14 h ON (blood alcohol levels during vapor exposure ranged between 150 and 250 mg%) and 10 h OFF, during which behavioral testing for acute withdrawal occurred (i.e., 6–8 h after the vapor was turned OFF, when brain and blood alcohol levels are negligible). In this model, rats exhibit somatic and motivational signs of withdrawal[41].

**Operant self-administration and withdrawal scores**. Behavioral testing occurred three times per week. The rats were tested for alcohol or saccharin (and water)

self-administration on an FR1 schedule of reinforcement in 30-min sessions. Behavioral signs of withdrawal were measured using a rating scale that was adapted from a previous study[42] and included ventromedial limb retraction (VLR), irritability to touch (vocalization), tail rigidity, abnormal gait, and body tremors. Each sign was given a score of 0–2, based on the following severity scale: 0 = no sign, 1 = moderate, 2 = severe. The sum of the four observation scores (0–8) was used as an operational measure of withdrawal severity.

**Intracranial surgery.** For intracranial surgery, the animals were anesthetized with isoflurane. For the behavioral experiments, a Cre-dependent adeno-associated virus that carried inhibitory opsins, AAV-DIO-eNpHR3.0-eYFP or AAV5-EF1a-DIO-eYFP (Vector Core, University of North Carolina, Chapel Hill, NC, USA), was injected bilaterally or unilaterally in the CeA (coordinates from bregma: anterior/posterior, −2.6 mm; medial/lateral, ± 4.2 mm; dorsal/ventral, −6.1 mm from skull surface). The injection was made through a stainless-steel injector that was 2 mm longer than the guide cannula (so that its tip protruded into the area) and connected to a 10 μl Hamilton syringe, which was controlled by an UltraMicroPump (WPI, Sarasota, FL, USA). Virus was injected (1.0 μl, 100 nl/min) over 10 min, followed by an additional 10 min to allow diffusion of the viral particles. For the rats that were destined for the behavioral experiments, chronic optical fibers were implanted bilaterally or unilaterally above the CeA (coordinates from bregma: anterior/posterior, −2.6 mm; medial/lateral, ± 4.2 mm; dorsal/ventral, −8.1 mm from skull surface) and unilaterally above the BNST (coordinates from bregma: anterior/posterior, −0.1 mm; medial/lateral, ± 1.4 mm; dorsal/ventral, −6.7 mm from skull surface), LH/pSTN (coordinates from bregma: anterior/posterior, −4.2 mm; medial/lateral, ± 1.2 mm; dorsal/ventral, −8.7 mm from skull surface), SI (coordinates from bregma: anterior/posterior, −0.1 mm; medial/lateral, ± 1.8 mm; dorsal/ventral, −8.5 mm from skull surface), and PBN (coordinates from bregma: anterior/posterior, −9.2 mm; medial/lateral, ± 1.8 mm; dorsal/ventral, −6.5 mm from skull surface). Following surgery, the rats were allowed to recover for 1 week.

**Viral vectors.** We used commercially available adenoviral constructs (Vector Core, University of North Carolina, Chapel Hill, NC, USA) for all of the experiments. All of the viruses were distributed into 10-μl aliquots, kept at −80°C, and thawed immediately before the injections.

**Locomotor activity.** Locomotor activity and anxiety-like behavior were evaluated in an opaque open field apparatus (100 cm × 100 cm × 40 cm) that was divided into 20 cm × 20 cm squares. On the experimental day, the animals were placed in the center of the open field, and locomotor activity (number of lines crossed), the time in the center, total distance traveled, and entries into the center were scored for 10 min. Behavior was videotaped, and AnyMaze software was used to analyze the recordings.

**Optical inhibition.** Optical probes were constructed, in which the optical fiber (200 μm core, multimode, 0.37 NA) was inserted and glued into a ceramic ferrule. The optical fiber and ferrule were then glued to a metallic cannula and closed with a dust cap. The optical fiber length was then adjusted based on the brain area of interest to not leave a gap between the skull and cannula during surgery. The other end of the optical fiber (FC/PC connection) was attached to a fiber splitter (2 × 1) that permitted simultaneous, bilateral illumination. The single end of the splitter was attached to a rotating optical commutator (Doric Lenses) to permit free movement of the rat. The commutator was connected to a fiber that was connected to a laser (DPSS, 200 or 300 mW, 532 nm, with a multimode fiber coupler for an FC/PC connection; OEM Laser Systems). Prior to the experiments, the light output of the optical fiber was adjusted to ~10 mW, measured by an optical power meter. Based on measurements in the mammalian brain[43] and assuming a geometric loss of light, a light output of 10 mW that is measured by a standard optical power at the tip of a fiber with an NA of 0.37 and fiber core radius of 200 μm will produce ~1 mW/mm² of light up to 1 mm directly away from the fiber tip, which is the minimum amount necessary to produce opsin activation[22,44]. Based on in vivo measurements of the shape of the light output in mammalian brain tissue, these parameters would be expected to provide sufficient light for opsin activation in at least 0.4 mm³ of tissue[45]. Optical fibers were implanted at the same time as the virus injections, and the animals were habituated to tethering to the commutator for several self-administration sessions before beginning the experiments. On the test days, the rats received optical inhibition paired with alcohol self-administration sessions, locomotor activity assessments, or withdrawal score assessments.

**Recruitment of CeA CRF neurons during alcohol withdrawal.** The animals were trained to self-administer 10% alcohol and made dependent on alcohol using the chronic intermittent exposure model. After the stabilization of excessive drinking, the animals were transcardially perfused. The brains were harvested, postfixed, and sectioned at 40 μm.

**Inhibition of CeA CRF neurons.** The animals were injected with AAV-DIO-eNpHR3.0-eYFP or AAV5-EF1a-DIO-eYFP and implanted with bilateral optical

fibers in the CeA. One week after surgery, the rats were trained to self-administer 10% alcohol until they reached a stable baseline of intake. At this point, the experiment began, and the rats were tested for alcohol self-administration during the optical inhibition of CeA CRF neurons. Sham optical inhibition sessions were performed before and after the test session. The animals were then placed in the alcohol vapor chambers and 3 weeks later underwent the alcohol self-administration escalation phase. When the escalation of alcohol intake was achieved, the animals were tested for the effects of the optical inhibition of CeA CRF neurons as described above. The effect of optical inhibition on somatic withdrawal signs was also assessed.

At the end of this phase, the rats were trained to self-administer saccharin for several sessions until a stable baseline of responses was reached. They were then tested for the effects of the optical inhibition of CeA CRF neurons on saccharin self-administration. The effects of optical inhibition on locomotor activity were also assessed.

**Dissection of the role of different CeA CRF pathways.** *Crh*-Cre rats were bilaterally infused with AAV5-EF1a-DIO-NpHR-eYFP in the CeA, unilaterally implanted with optical fibers in the CeA, BNST, PBN, SI, and LH, and made dependent on alcohol using the chronic intermittent exposure model. After the escalation of drinking, the animals were tested for the effect of the optogenetic inhibition of CeA CRF projections on alcohol drinking, alcohol withdrawal signs, and saccharin self-administration.

**Immunohistochemistry.** *Crh*-Cre rats were implanted with unilateral optical fibers in the CeA and tested for the effect of inhibition of the CeA for 30 min during withdrawal (8 h) from chronic intermittent alcohol. Ninety minutes later, the animals were deeply anesthetized and perfused with 100 ml of phosphate-buffered saline (PBS) followed by 400 ml of 4% paraformaldehyde. The brains were post-fixed in paraformaldehyde overnight and transferred to 30% sucrose in PBS/0.1% azide solution at 4°C for 2–3 days. Brains were frozen in powdered dry ice and sectioned on a cryostat. Coronal sections were cut 40 μm thick between bregma + 4.2 and −6.48 mm[46] and collected free-floating in PBS/0.1% azide solution. Following three washes in PBS, the sections were incubated in 1% hydrogen peroxide/PBS to quench endogenous peroxidase activity, rinsed three times in PBS, and blocked for 60 min in PBS that contained 0.3% TritonX-100, 1 mg/ml bovine serum albumin, and 5% normal donkey serum. The sections were incubated for 24 h at 4°C with rabbit monoclonal anti-Fos antibody (Cell Signaling Technology, catalog no. 2250) diluted 1:1000 in PBS/0.5% Tween-20 and 5% normal donkey serum. The sections were washed again with PBS and incubated for 1 h in undiluted Rabbit ImmPress HRP reagent (Vector Laboratories). After washing in PBS, the sections were developed for 2–6 min in Vector peroxidase DAB substrate (Vector Laboratories) enhanced with nickel chloride. Following PBS rinses, the sections were mounted on coated slides (Fisher Super Frost Plus), air dried, dehydrated through a graded series of alcohol, cleared with Citrasolv (Fisher Scientific), and coverslipped with DPX (Sigma).

Quantitative analysis to obtain unbiased estimates of the total number of Fos+ cell bodies was performed using a Zeiss Axiophot Microscope equipped with MicroBrightField Stereo Investigator software (Colchester, VT, USA), a three-axis Mac 5000 motorized stage (Ludl Electronics Products, Hawthorne, NY, USA), a Q Imaging Retiga 2000R color digital camera, and a PCI color frame grabber.

**Double-labeled immunohistochemistry.** Immunohistochemistry was used to characterize the type (CRF) and number (NeuN) of neurons that were activated (Fos+) during alcohol withdrawal. Coronal sections were cut 40 μm thick between bregma −2.0 and −5.0 mm[46]. We determined the proportion of all neurons that expressed Fos during alcohol withdrawal by double-labeling for Fos and the neuron-specific protein NeuN and the population of activated CRF neurons in the CeA by double labeling for Fos and CRF.

For Fos/NeuN and Fos/CRF double-labeling, 40 μm sections were washed three times in PBS and permeabilized/blocked for 60 min in PBS with 0.2% Triton X-100, 5% normal donkey serum, and 3% bovine serum albumin (blocking solution). The sections were incubated in primary antibodies diluted in blocking solution for 24 h on a shaker at room temperature. Primary antibodies were used at the following concentrations: anti-Fos (1:500, Millipore, catalog no. AB4532) and anti-NeuN (1:1000, Millipore, catalog no. MAB377). The sections were washed three times in PBS for 10 min and incubated with fluorescently labeled secondary antibodies diluted in PBS for 2 h on a shaker at room temperature. The secondary antibodies were Alexa Fluor 488-labeled donkey anti-rabbit (1:500, Invitrogen, catalog no. A-10042), Alexa Fluor 568-labeled donkey anti-mouse (1:500, Invitrogen, catalog no. S-11249), and Alexa Fluor 647-labeled donkey anti-goat (1:500, Invitrogen, catalog no. A-21447). After labeling, the sections were washed three times in PBS for 10 min, mounted on Fisher Superfrost Plus slides (catalog no. 12-550-15), and coverslipped with PVA-Dabco (Sigma).

We quantified Fos+/CRF+ expression when a neuron exhibited a Fos staining pattern as a focused nucleus (red) and a cytoplasm that was stained for CRF (green). Three sections from each rat were bilaterally quantified within the boundary of the CeA and normalized by area, measured by Fiji software. Counts from all images for each rat were averaged so that each rat was an *n* of 1. For DAB

staining, brown (NeuN) and black (Fos) channels were color-separated using Fiji software to isolate the red channel. Using the red channel as the background channel for thresholding leaves dark nuclei from the black (Fos) channel easy to identify and threshold. Thresholding was applied under the same conditions for both naive and withdrawal. Thresholded images were counted for positive foci within the nucleus of NeuN-positive neurons.

**Confocal acquisition and three-dimensional analysis.** Three-dimensional stacks of images were acquired with a 780 Laser Scanning Confocal microscope (Zeiss) using a 20× (1 μm image slice), 40× (0.6 μm image slice), or 63× (0.2 μm image slice) objective to observe the entirety of the CeA. The system is equipped with a stitching stage and Zen software to reintegrate the tiled image stacks. Stitched z-series images of the entire CeA were imported into Imaris software (Bitplane-Andor) and Fiji for quantification.

**Slice preparation for whole-cell recordings.** $Crh$-Cre rats ($n = 20$) that received bilateral infusions of rAAV5/Ef1a-DIO eNpHR in the CeA were deeply anesthetized with 3% isoflurane 4–5 weeks after virus infusion and transcardially perfused with ice-cold oxygenated sucrose solution. The rats were then decapitated. Brains were rapidly removed and placed in oxygenated (95% $O_2$, 5% $CO_2$) ice-cold cutting solution that contained 206 mM sucrose, 2.5 mM KCl, 1.2 mM $NaH_2PO_4$, 7 mM $MgCl_2$, 0.5 mM $CaCl_2$, 26 mM $NaHCO_3$, 5 mM glucose, and 5 mM HEPES. Slices of the CeA or BNST (300 μm) were cut on a VT1000S Vibratome (Leica Microsystems, Buffalo Grove, IL, USA) and transferred to oxygenated artificial cerebrospinal fluid (aCSF) that contained 130 mM NaCl, 2.5 mM KCl, 1.25 mM $NaH_2PO_4$, 1.5 mM $MgSO_4 \cdot 7H_2O$, 2.0 mM $CaCl_2$, 24 mM $NaHCO_3$, and 10 mM glucose. The slices were first incubated for 30 min at 37℃ and then kept at room temperature for the remainder of the experiment. Individual slices were transferred to a recording chamber that was mounted on the stage of an upright microscope (Olympus BX50WI, Waltham, MA, USA) and continuously perfused in oxygenated aCSF at a rate of 2–3 ml/min. Neurons were visualized with a 60× water immersion objective (Olympus), infrared differential interference contrast optics, and charge-coupled device camera (EXi Blue, QImaging, Surrey, Canada). Whole-cell recordings were performed with a Multiclamp 700B amplifier, Digidata 1440 A, and pClamp 10 software (Molecular Devices, Sunnyvale, CA, USA). Patch pipettes (4-7 MΩ) were pulled from borosilicate glass (Warner Instruments, Hamden, CT, USA) and filled with the following internal solution: 70 mM $KMeSO_4$, 55 mM KCl, 10 mM NaCl, 2 mM $MgCl_2$, 10 mM HEPES, 2 mM Mg-ATP, and 0.2 mM Na-GTP.

To elicit neuronal inhibition, green light (532 nm) was turned ON for 6.5 s or 5 min via a Master-8 stimulator (AMPI, Jerusalem, Israel) that generated a light train at 15 Hz. Neurons that expressed eNpHR within the CeA were visualized with differential interference contrast and widefield fluorescence imaging using an Olympus 60× immersion objective. To identify eNpHR-expressing neurons, a Lambda DG-4 light source was used with an in-line EX540/EM630 filter set. A Mosaic 3 pattern illuminator (Andor Instruments, Belfast, UK) coupled to a 532 nm light-emitting diode (CoolLED Limited, Andover, UK) was attached to the microscope and used for light delivery (10 mW/mm$^2$, to approximate output in the behavioral experiments) through the objective within the slice preparation.

**Histology.** Following completion of the behavioral experiments, the rats were deeply anesthetized and transcardially perfused. Brains were removed, and 40 μm coronal slices that contained the CeA, BNST, SI, LH-pSTN, and PBN were cut on a cryostat. The brain slices were mounted on microscope slides, and the expression of viral vectors and/or optical fiber placements were examined for all of the rats using fluorescent microscopy. Rats that had no YFP expression in the CeA because of faulty microinjections or had fiber placements that failed to target the regions of interest were excluded from the behavioral analysis.

**Statistical analysis.** The data are expressed as mean ± SEM. For comparisons between only two groups, $p$ values were calculated using paired or unpaired $t$-tests as described in the figure legends. Comparisons across more than two groups were made using one-way analysis of variance (ANOVA), and two-way ANOVA was used when there was more than one independent variable. Data from experiments that tested the effect of the unilateral optogenetic inhibition of CeA CRF neurons on the escalation of alcohol self-administration in dependent rats were analyzed using two-way repeated-measures ANOVA, with time and laser illumination as within-subjects factors (see Figs. 4 and 5, Supplementary Figures 4, 5, 6). In the same experiments, differences between baseline intake before dependence and intake after vapor exposure (ON and OFF days; see Figs. 4 and 5, Supplementary Figures 4, 5, 6) were analyzed using separate one-way repeated-measures ANO-VAs. Significant effects in the ANOVA were followed by the Newman Keuls $post$ $hoc$ test. Withdrawal signs were analyzed using the nonparametric Mann-Whitney $U$ statistic, followed by Dunn's multiple-comparison test. The standard error of the mean is indicated by error bars for each group of data. Differences were considered significant at $p < 0.05$. All of these data were analyzed using Statistica 7 software.

**Reporting summary.** Further information on experimental design is available in the Nature Research Reporting Summary linked to this article.

## Data availability

The datasets that were generated and/or analyzed during the present study are available from the corresponding author upon reasonable request.

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

## Acknowledgements

We thank Michael Arends for proofreading the manuscript. This study was supported by National Institutes of Health grants AA006420, AA020608, and AA022977 to O.G. and AA13588 and AA026075 to R.O.M.

## Author contributions

G.d.G. and O.G. were responsible for the study concept and design. GdG contributed to the acquisition of animal data. M.B.K. and P.S. contributed to the acquisition of electrophysiology data. G.d.G., M.B.P., E.C., and S.S. contributed to the immunohistochemistry experiments. ROM helped with the experimental design and provided the *Crh*-Cre rats. G.d.G., R.O.M., G.F.K., and O.G. assisted with the data analysis and interpretation of the data and drafted the manuscript. G.F.K. and R.O.M. provided critical revision of the manuscript. All of the authors critically reviewed the content and approved the final version for publication.

## Additional information

**Competing interests:** The authors declare no competing interests.

