## [Peer Review File · Nature Communications]

Reviewers' Comments:

Reviewer #1:

Remarks to the Author:

This manuscript presents experiments aimed at determining the role of neuronal ensembles in the central nucleus of the Amygdala (CeA) and associated downstream neural pathways in mediating aversive aspects of withdrawal from alcohol (EtOH), and in particular their role in escalated alcohol consumption as a form of presumed negative reinforcement. The authors use immunocytochemistry and optogenetics to identify the corticotropin releasing hormone (Crh) subset of CeA neurons and their connection specifically to the bed nucleus of the stria terminalis (BNST) as primary mediators of EtOH withdrawal symptoms and associated escalation of EtOH consumption. Specifically, the authors demonstrate that Crh containing CeA neurons make up the majority of activated CeA neurons (as assessed by immunostaining for the immediate early gene c-Fos) during EtOH withdrawal, and that selective optogenetic inhibition of such Crh neurons during EtOH withdrawal reduces CeA activation (both Crh and non-Crh neurons) and withdrawal symptoms, and, amazingly, completely eliminates escalated EtOH consumption. Subsequent selective optogenetic inhibition of Crh-positive CeA axon terminals in various efferent nuclei indicated that the BNST is the only output pathway to mediate such actions, and it does so to the same extent as CeA inhibition. All such impacts appear to be specific to EtOH withdrawal and withdrawal-induced escalated EtOH consumption, as evidenced by a variety of control experiments showing no impacts of optogenetic suppression of CeA and/or CeA to BNST afferents on basal EtOH consumption, water consumption, sucrose consumption, or locomotor activity.

Based on these observations the authors conclude that the Crh containing neuronal ensemble within the CeA is the primary mediator of CeA activation, and via its connections to the BNST is the primary mediator of associated aversive withdrawal symptoms and consequent escalation of EtOH consumption.

The topic of this manuscript is obviously of high social and clinical relevance, with alcohol abuse being a major burden on the health and socioeconomic fabric of our society. And, the social implications combined with the techniques used and circuitry mapped, should make this manuscript of interest to the wider readership. The rationale for the project, the approach and techniques are all very logically and clearly presented, and the specific experiments are, for the most part, well designed, executed, and presented. The authors have also emphasized implementation of a variety of important controls, in particular with respect to the use of optogenetics, which is important. Overall this is a very elegant study.

There are, however, a few issues with the execution and presentation of several of the specific experiments, including control experiments (see specific comments for details). Briefly though, the methodology for quantifying the various immunocytochemistry studies is not very well described, and it is difficult, as presented, to be confident that the mean data accurately reflects the raw data and presumed meaning of the raw data. Further, although the authors have emphasized their implementation of important control experiments for the optogenetic studies, which is commendable, there are some problems with the way they have been implemented, and key controls have not been done for the afferent studies, and that lack seriously compromises interpretation of the in vivo data. These issues should be addressed.

Lastly, the results of this study are actually quite remarkable in how a single, arguably minor pathway, apparently completely mediates what is arguably a very complex scenario: EtOH withdrawal and associated escalated EtOH consumption. Consequently, further exploration and discussion of how this is possible is warranted (see specific comments for details).

Issues:

1) Although the behavioral data for the studies of HaloR inhibition of CeA afferents to various nuclei is quite clear and striking (Fig. 5), important controls have not been done, and this seriously hampers overall interpretation of the data. Specifically, the authors need to test whether light activation of HaloR in specific afferent terminal targets actually inhibits axon terminals enough to significantly reduce synaptic transmission. This needs to be tested with slice studies, and ideally in vivo c-Fos studies. Such important controls need to be done in the proposed effective path (CeA to BNST), but especially in the non-effective pathways. It is difficult to inhibit action potential invasion of synaptic terminals, and hence vesicle release, with a small (5mV) hyperpolarization (and that assumes the hyperpolarization seen at the terminals is of the same magnitude at that observed at the soma, Fig. 2C&D?). This is important because although the behavioral data make a strong case for the importance of the CeA to BNST pathway, the conclusion that it is the only relevant pathway is completely dependent on assuming the light stimulation of HaloR in the other pathways is effective at inhibiting their target cell's excitation, which may not be the case, even if it is in the BNST afferents.

The authors should also provide some quantification of the estimated percent inhibition of the various nuclei tested (i.e. is the lack of effect in other nuclei simply due to the possibility that a smaller percent of total relevant neurons are inhibited). Such estimation could be based on volume of nuclei relative to estimated light volume, or better yet with actual quantification of percent silencing of c-Fos neurons in each nuclei.

2) Related to the comment above, the most striking result of this study is that light activated HaloR in the CeA to BNST terminals (unilaterally) is enough to completely inhibit withdrawal-induced escalated EtOH consumption (Fig. 5B), despite having no impact on three of the 5 withdrawal symptoms tested (Fig. 5C; and unknown impact on other withdrawal symptoms). This raises the question of how such inhibition is causing such an outcome? The way the manuscript is introduced, one might infer that the mechanism is envisioned to be reduced withdrawal symptoms, and thus less drive for EtOH consumption as a negative reinforcer. However, this does not seem to fit with the fact that many withdrawal symptoms that are presumably aversive persist (Fig. 5C). An alternative possibility that is not considered is that the circuit plays a role in associative learning, in a manner that inhibition of the circuit inhibits memory of the ameliorative aspects of EtOH, which might then be expected to eliminate escalated EtOH consumption despite persistence of various aversive symptoms. It would increase the impact of this manuscript a great deal if the authors could discriminate between these two possibilities. Another possibility is that EtOH itself only reduces some of the aversive withdrawal symptoms, most parsimoniously the ones that are eliminated when CeA to BNST afferents are inhibited (abnormal gait & body tremors), and so optogenetic inhibition effectively provides the same relief as EtOH, both being incomplete but equally reinforcing. It would be informative to determine what EtOH does to the tested withdrawal symptoms.

3) A final concern, related to comments 1&2 above, is the fact that unilateral inhibition of the Crh CeA neurons or their afferents to the BNST completely prevents escalated EtOH consumption (Fig. 4B). This is somewhat surprising. Specifically, it is difficult to understand this complete block of escalated consumption in the context of the overall proposed model, i.e. that CIE-induced altered Crh-CeA signaling underlies the aversive aspects of withdrawal that drive escalated EtOH consumption as a negative reinforcer. Do the authors propose that one hemisphere of such alterations has no aversive impact? Perhaps relatedly, in Fig. 4H it appears that the image of the CeA is a mirror image orientation of the images in Fig4F&G (based on the position of the CeA relative to the direction of the tip of the nearby fissure). Is this the contralateral hemisphere? Either way it would be informative to examine C-Fos expression in the contralateral amygdala. Does unilateral inhibition somehow inhibit both hemispheres? If not, the authors should at least discuss this issue.

4) The authors quantify EtOH consumption by “number of rewards” (presumably the number of active lever presses) during Operant self-administration trials. While this is an understandably easily quantified measurement, it does not inform us of the blood EtOH concentrations (BECs) achieved during such trials, which has several important ramifications and/or drawbacks. First, it does not seem appropriate to refer to such measurements as “excessive drinking”, page 4, 3rd sentence. All we can say is that drinking increased. Second, and more importantly, in the context of the EtOH consumption and escalation during EtOH withdrawal, it is implied that the driving force for elevated consumption is amelioration of the associated aversive state. Thus it is important to determine whether BECs achieved are actually high enough to significantly alter CeA signaling. Such information might partially inform issue #2 above. Additionally, it is important to know what potential molecular mediators of such action are, which requires knowing the BECs achieved. I.e. are the levels achieved likely able to suppress CeA NMDARs, enhance GABAARs, alter transmitter release, alter Crh, etc. Thus for both aspects, it is important to determine and present BECs achieved during baseline and elevated EtOH consumption.

5) There are a variety of issues that make it difficult to assess the outcomes of the immunocytochemical analysis and the relationship between the raw data and mean data as presented in the bar graphs (Figs 1E&F and Suppl. Fig. 1 C&D, and Fig. 2B). In particular, there is a considerable amount of ambiguity about the nature and magnitude of the staining to the naked eye in the raw data images, and there is an inadequate description of the quantitative details of how bar graphs were generated from such ambiguous images. There is also a lack of controls showing staining is specific and/or represents what the authors express in the bar graphs. Of specific concern in Fig. 1E&F is how the authors determined what counted as a c-Fos positive or negative cell. First, proper immunocytochemical controls showing that staining is specific to c-Fos are needed, or at least a citation to prior work showing that the specific antibody used is specific. Second, while the arrows appear to point to what the authors conclude are c-Fos positive cells (note, although deducible, the legend lacks a description of what the arrows and their color signify), there is clearly abundant green cytoplasmic staining in a much larger proportion of the exhibited cells but with a different apparent morphology (apparent cytoplasmic rings versus solid circles of color). What are the respective staining patterns? Is this an artifact of confocality (with rings versus solid circles depending on depth into a given cell)? Is the antibody specific? If the antibody is specific, what are the apparent cytoplasmic rings, and do they change (from the exemplar images it looks like rings disappear in withdrawal animals, does this reflect redistribution of c-Fos?), and what does this mean? Similarly, in Supplemental Fig. 1 C&D, while one can get a vague sense that arrows point to “darker” spots, it is not by any means clear. Basically every nucleus has some degree of brownish/black, and there is no description of how thresholding was determined or implemented, or how the accuracy of such implementation was confirmed? Furthermore, the dark spots do not appear to co-localize with the blue stain (presumed NeuN, indicative of neuronal subtype), which is supposed to indicate neuronal overlap. Again, a better description of methodology and confirmation of validity is needed.

6) There are some important issues with the electrophysiological analysis of the impact of HaloR activation on CeA neuronal excitability. First, the language describing the studies is confusing. The recordings are clearly in the current-clamp configuration (i.e. clamping electrode current to record membrane potential), but the authors refer to holding potential (last sentences on page 5) which implies voltage-clamp (i.e. clamping membrane potential to record currents). This language needs to be fixed. More importantly, there is no description of what the resting membrane potential was (and whether it was similar across conditions), or how much current was injected to alter the membrane potential to what one presumes was designed to be similar across cells? Most importantly though, this being a crucial control experiment, as best I can tell the authors have injected current to adjust the membrane potential to be right above action potential (AP) threshold, hence the steady action

potential firing. While this approach provides some information (i.e. it clearly shows that light excitation of HaloR slightly hyperpolarizes CeA cells, and if they are just above AP threshold, it can silence their AP firing), it does not tell us much about the likely impact of light-activation of HaloR under normal physiological conditions, when cells are not manipulated, by constant current injection, to be just above AP threshold. What is more relevant is what impact such a small hyperpolarization has on post synaptic AP responses to signals from other cells, either in the form of glutamatergic/GABAergic or CRF transmission. This is particularly important given the very small hyperpolarization (5mV) induced by light activation of HaloR. The authors should determine whether light excitation of HaloR can reduce AP output induced by afferent stimulation or CRF application, when the cells are not artificially depolarized with current injection. A simpler but less elegant alternative, would be to examine the impact of light inhibition on responses to transient current injections from a cell at its native resting potential. Also, in both the existing experiment and any future experiments, the authors should provide detailed quantification of the change in AP output, as opposed to just stating that it "suppressed AP firing rate" (line 95). I emphasize, combined with the behavioral data I don't doubt that the HaloR stimulation can reduce CeA firing in vivo, but it is important to know more realistically how such behavioral changes are induced. As the figure stands it might be concluded that HaloR stimulation completely shuts down CeA firing, which is highly unlikely to be the case under more physiological circumstances. This is important for our understanding of how much cellular inhibition is required to implement such dramatic behavioral outcomes.

7) The graphical display of withdrawal symptom subsets is too small (Figs. 4C and 5C-L). While I appreciate the value of the combined score insert, the potentially more informative comparisons of withdrawal subset comparisons across Crf-CeA inhibition and CeA to BNST afferent inhibition is very difficult to see given the scaling required to accommodate the insert. The Y-axis scale of the subset graphs should be expanded so that comparisons across the two conditions can be more easily made. The large size of combined totals insert, while informative, obscures differences in the subcategories. In this regard, it appears that the only subcategory that is significantly reduced by both optical inhibition of the Crf-CeA and CeA to BNST afferents is "abnormal gait", despite both completely inhibiting escalated EtOH consumption. Does this imply abnormal gate is the main behavioral manifestation of EtOH withdrawal that drives escalation of EtOH consumption? If so, how does this relate to the known functions of CeA to BNST circuit, and/or downstream aspects of the circuit?

Minor comments:

1) The authors need to refer to a manuscript that describes the generation and characterization of the Crh-Cre rats used in this study. Apologies if I missed it somewhere, but it is not in the first reference to the animals or in the methods that describe their use as subjects.

2) The authors should explicitly state the precise time frame (relative to CIE termination) for when operant self-administration and withdrawal symptoms were quantified.

Reviewer #2:

Remarks to the Author:

The manuscript by George and colleagues seeks to assess the role for neuronal activations of the central nucleus of the amygdala (CeA) aspects of alcohol behaviors that are associated with dependence/compulsive drinking in rodent models. The overall hypothesis was that corticotropin-releasing factor (CRF) neurons and downstream targets, such as their known projections to the bed nucleus of the stria terminalis (BNST), might be involved and that these CRF neurons and related circuitry might be a potential target for reducing compulsive drinking in AUD. These authors have utilized a number of state-of-the-art techniques to test their hypothesis, and have provided a well

written and compelling study. The hypothesis that the CeA is involved in AUD is solidly based on a large pre-clinical literature, notably that neuronal ensembles in the the CeA are associated with alcohol withdrawal behaviors in alcohol-dependent rats. A large body of evidence implicates CRF neurotransmission in the CRF in withdrawal and negative emotional states, and supportive pharmacological evidence for a CRF CeA link.

In this study, first Crh-Cre transgenic rats were used in an animal model of alcohol dependence combined with in vivo optogenetics and immediate early gene brain mapping. Then the role for several possible downstream CeA-CRF projections were assessed by optogenetic inactivation methods, i.e., CRF terminals from CeA neurons that project to the BNST (CRFCeA-BNST), LH and pSTN (CRFCeA-LH/pSTN), substantia innominata (SI; CRFCeA-SI), and parabrachial nucleus (PBN; 59 CRFCeA-PBN).

The main finding of these experiments were that 1) withdrawal from alcohol dependence significantly and mostly affects CRF neurons (~ 80%), 2) that the inactivation of these CeA CRF+ neurons prevents the recruitment of the CeA neuronal ensemble, and reduces dependence-associated alcohol drinking and withdrawal, and 3) that based on the optogenetic circuitry assessment that it is activation of CeA neurons that project to the BNST that are critically involved.

The paper is exceptionally well prepared and generally presented. These are several issues of interpretation and some details that need to be addressed.

The conclusion that CRF neurons are critical to “drive compulsive-like alcohol drinking” is unclear. Although dependent rats were found to have a significant proportion of the CeA neuronal ensemble that is associated with withdrawal in alcohol-dependent rats, it seems a stretch to argue that this reflects compulsive (or compulsive-like) behavior, or what component of “compulsion”.

While these data do seem to support the overall conclusion that there are key CeA neuronal ensemble that are involved in alcohol drinking in dependent rats – it is not clear that this is “excessive” nor “compulsive” nor is clear if this is only a result from the vapor-inhalation model. These data do nonetheless show that inactivation of the CRFCeA-BNST pathway – rather selectively -- could reverse this “excessive” alcohol drinking during withdrawal and partially prevented the somatic signs of withdrawal. More discussion and validation is needed to argue that these data have identified a cellular mechanism of excessive alcohol use, and it appears that with regards to this statement in the Introduction they cite their own JN paper. Indeed, in the Results the word “escalation” of intake is used to describe these actual data (Figure 1A), though then at other points it is then referred to as “excessive”, and then as noted above, the term “compulsive” then gets introduced when these -- though the title and one place in the Discussion uses the careful “addiction-like” term, even though the Discussion starts with “compulsive-like”. I think that words matter.

The details with regards to numbers of animals used for the statistical analyses are not transparent in several places. In fact (perhaps I missed it) I cannot find any details with regards to number of animals for each experiment/condition with the exception of the whole-cell recordings. Statistical analyses section is too general and does not make it clear the numbers of animals that were used/reported in these data presented in the figure legends.

A number of minor issues need to be addressed.

It would be helpful that scale the axis for comparisons to be matched. For example in Figure 5 the axis showing the rewards with the optogenetic inhibition of the inhibition BNST is not the same as for the SI, LH, and PBN for the alcohol SA data.

There is no justification for the exclusive use on male rats, and given NIH guidelines female rats should be included.

There are scant details with regards to the operant self-administration and methods used to “get” animals to drink the 10% (w/v) alcohol, and for the saccharin self-administration study, and whether the alcohol animals were initially trained to drink saccharin and alcohol in a “fading” design.

These studies are based on a model of alcohol dependence produced by exposure to chronic intermittent alcohol vapor. While many groups use this model, it is not clear to the extent this model has clear translational relevance and that these effects can be ascribed selectively to alcohol pharmacological effects as opposed to more general effects of stress or aversion. For example, while the number of Fos+/CRF+ neurons in the CeA significantly increased during withdrawal this is based on the comparison between alcohol-withdrawn and alcohol-naïve rats – and in my opinion the alcohol-naïve rats are not an appropriate control.

Given the above concern, the authors could explore other alcohol drinking models other than vapor inhalation.

These whole-cell patch electrophysiological data to confirm the efficacy of NpHR in preventing CRF+ neuron firing in acute CeA slices from Crh-Cre rats was not well described or integrated into the manuscript.

It is confusing that the Figures with legends are presented before the References cited, and then shown again without the legends at the end of the manuscript.

Reviewer #3:

Remarks to the Author:

Major Concerns:

The authors examine many potential outputs of CeA CRF neurons including the Bed Nucleus of the Stria Terminalis (BNST), Substantia Inominata (SI), Lateral Hypothalamus (LH), and Parabrachial Nucleus (PBN). While some of these outputs are well-known, such as the BNST and PBN, others, such as the SI and LH, have not been described previously. Given light microscopy cannot differentiate between terminals and fibers of passage, it is unclear whether the lack of effects seen in these regions are true effects or are simply without effect due to the lack of synapses present. The authors must test the function of these outputs in order to claim that inhibition of these outputs are without effect. One potential way to resolve this question would be to perform light-evoked recordings in this region and examine what percentage of cells are responsive to stimulation.

Going further, what is the overlap between BNST-projecting CRF neurons and the other outputs tested? Are BNST projections a separate population or are they collaterals of the same cells?

The authors find that both somatic inhibition and inhibition of BNST outputs produce similar effects. This does not preclude a role for local CRF neurons in the CeA in regulating alcohol intake. Given the body of work describing the effects of alcohol on CRFR1+ cells in the CeA (Herman et al. 2013; Herman et al. 2016), it would be helpful to examine a role for CRF neurons in the CeA using an intersectional strategy to target non-BNST projecting neurons.

A major limitation of this paper is lack of any evidence describing a role for CRF signaling itself. The authors acknowledge that CeA CRF neurons also express a medley of other neuropeptides such as dynorphin, neotensin, and somatostatin, all of which may be contributing to the effects seen. Thus it

is unclear whether the results seen are due to manipulation of CRF signaling or inhibition of a unique subset of CeA neurons.

Another major limitation of this study is the lack of appropriate non-dependent controls. Studies using CIE should include air exposed controls to show that effects are due to induction of alcohol-dependence. More seriously, there are no virus controls for the experiments in figure 5. Without these data points, it is not possible to determine whether the experiments in this aim are the result of a true effect of inhibition of the pathway or a spurious result of laser stimulation.

The authors claim they targeted CRF neurons in the CeL, however there is also a population of CRF neurons in the CeM. Given the lack of clear anatomical boundaries between these two subdivisions, it is difficult to believe that expression was constrained to the CeL.

Throughout the manuscript statistics are poorly described, and in several cases do not match the data shown.

The authors do not demonstrate effective presynaptic inhibition using the in vivo parameters. This is a topic of great debate and needs to be included.

Minor Concerns:

For introduction cite original research papers rather than reviews?

What quantity of alcohol is administered by the rats? All the figures show the number of rewards obtained, however it would be more informative to report this as grams of ethanol per kilogram of body weight. This allows the readers to understand the effective dose of alcohol animals are receiving and would normalize differences in body weight across animals.

The Whole-Cell Patch Clamp recording section begins rather abruptly. The manuscript would read more smoothly if a few sentences of introduction were added stating that the purpose of these experiments were to validate the efficacy of DIO-NpHR in the CeA of CRF-Cre rats. Likewise, it would be helpful to have a more descriptive title such as "Halorhodopsin effectively inhibits CeA CRF neurons" rather than just calling it "Whole-Cell Patch Clamp Recording".

In the methods section, a specific age range of animals should be given rather than listing them as "adults".

We thank the reviewers for their very detailed and thoughtful comments on the original manuscript. As suggested, we performed a large number of additional experiments to address their concerns, including (1) testing behavioral controls using animals that were not injected with virus, (2) testing the effect of terminal inhibition using slice electrophysiology, (3) providing additional data analysis for the neuroanatomical characterization of Fos, (4) testing the effect of unilateral inhibition of the CeA on bilateral Fos counts, and (5) further characterizing the effect of the optogenetic inhibition of CRF neurons on action potentials. We believe the main conclusions of the revised manuscript (role of CRF^{CeA-BNST} neurons in alcohol drinking in dependent animals) were considerably strengthened by these additional experiments. We hope the revised manuscript addresses the reviewers' major concerns.

Reviewer 1

This manuscript presents experiments aimed at determining the role of neuronal ensembles in the central nucleus of the Amygdala (CeA) and associated downstream neural pathways in mediating aversive aspects of withdrawal from alcohol (EtOH), and in particular their role in escalated alcohol consumption as a form of presumed negative reinforcement. The authors use immunocytochemistry and optogenetics to identify the corticotropin releasing hormone (Crh) subset of CeA neurons and their connection specifically to the bed nucleus of the stria terminalis (BNST) as primary mediators of EtOH withdrawal symptoms and associated escalation of EtOH consumption. Specifically, the authors demonstrate that Crh containing CeA neurons make up the majority of activated CeA neurons (as assessed by immunostaining for the immediate early gene c-Fos) during EtOH withdrawal, and that selective optogenetic inhibition of such Crh neurons during EtOH withdrawal reduces CeA activation (both Crh and non-Crh neurons) and withdrawal symptoms, and, amazingly, completely eliminates escalated EtOH consumption. Subsequent selective optogenetic inhibition of Crh-positive CeA axon terminals in various efferent nuclei indicated that the BNST is the only output pathway to mediate such actions, and it does so to the same extent as CeA inhibition. All such impacts appear to be specific to EtOH withdrawal and withdrawal-induced escalated EtOH consumption, as evidenced by a variety of control experiments showing no impacts of optogenetic suppression of CeA and/or CeA to BNST afferents on basal EtOH consumption, water consumption, sucrose consumption, or locomotor activity. Based on these observations the authors conclude that the Crh containing neuronal ensemble within the CeA is the primary mediator of CeA activation, and via its connections to the BNST is the primary mediator of associated aversive withdrawal symptoms and consequent escalation of EtOH consumption. The topic of this manuscript is obviously of high social and clinical relevance, with alcohol abuse being a major burden on the health and socioeconomic fabric of our society. And, the social implications combined with the techniques used and circuitry mapped, should make this manuscript of interest to the wider readership. The rationale for the project, the approach and techniques are all very logically and clearly presented, and the specific experiments are, for the most part, well designed, executed, and presented. The authors have also emphasized implementation of a variety of important controls, in particular with respect to the use of optogenetics, which is important. Overall this is a very elegant study. There are, however, a few issues with the execution and presentation of several of the specific experiments, including control experiments (see specific comments for details). Briefly though, the methodology for quantifying the various immunocytochemistry studies is not very well described, and it is difficult, as presented, to be confident that the mean data accurately

reflects the raw data and presumed meaning of the raw data. Further, although the authors have emphasized their implementation of important control experiments for the optogenetic studies, which is commendable, there are some problems with the way they have been implemented, and key controls have not been done for the afferent studies, and that lack seriously compromises interpretation of the in vivo data. These issues should be addressed. Lastly, the results of this study are actually quite remarkable in how a single, arguably minor pathway, apparently completely mediates what is arguably a very complex scenario: EtOH withdrawal and associated escalated EtOH consumption. Consequently, further exploration and discussion of how this is possible is warranted (see specific comments for details).

Response: We thank the reviewer for the positive evaluation of the manuscript. We performed numerous additional experiments and revised the manuscript accordingly to address the concerns.

Issues:

1) Although the behavioral data for the studies of HaloR inhibition of CeA afferents to various nuclei is quite clear and striking (Fig. 5), important controls have not been done, and this seriously hampers overall interpretation of the data. Specifically, the authors need to test whether light activation of HaloR in specific afferent terminal targets actually inhibits axon terminals enough to significantly reduce synaptic transmission. This needs to be tested with slice studies, and ideally in vivo c-Fos studies. Such important controls need to be done in the proposed effective path (CeA to BNST), but especially in the noneffective pathways. It is difficult to inhibit action potential invasion of synaptic terminals, and hence vesicle release, with a small (5mV) hyperpolarization (and that assumes the hyperpolarization seen at the terminals is of the same magnitude at that observed at the soma, Fig. 2C&D?). This is important because although the behavioral data make a strong case for the importance of the CeA to BNST pathway, the conclusion that it is the only relevant pathway is completely dependent on assuming the light stimulation of HaloR in the other pathways is effective at inhibiting their target cell's excitation, which may not be the case, even if it is in the BNST afferents.

The authors should also provide some quantification of the estimated percent inhibition of the various nuclei tested (i.e. is the lack of effect in other nuclei simply due to the possibility that a smaller percent of total relevant neurons are inhibited). Such estimation could be based on volume of nuclei relative to estimated light volume, or better yet with actual quantification of percent silencing of c-Fos neurons in each nuclei.

Response: To address this important comment, we performed additional slice experiments. The results are presented in Fig. 5, showing that the inhibition of axon terminals in the BNST is sufficient to reduce neuronal activity, confirming the behavioral data. We revised the Discussion to read, "In slice experiments, we found that the inhibition of CeA CRF axon terminals in the BNST reduced the firing of a subpopulation of BNST neurons. This result is consistent with previous studies that reported that CRF can enhance spontaneous excitatory transmission in the BNST through an increase in glutamate release (Kash et al., 2008; Walker and Davis, 2008). A similar potentiation of excitatory transmission by CRF has been reported in the CeA and the lateral septum mediolateral nucleus (Liu et al., 2004)."

Although we were able to test the effect of the inhibition of axon terminals in the CRF^{CeA-BNST} projection using slice electrophysiology and confirm the validity of the optogenetic approach, we were unable to perform these experiments for all of the projections (i.e., CeA to LH, PBN, and SI). It took us ~9 months to perform the additional five experiments that are reported in the current revision of the manuscript. These remaining additional experiments are quite laborious and would require another 8-9 months of work (at best), and we would need to breed several new cohorts of animals (5 months) to obtain at least 30-40 animals, perform stereotaxic surgery (2 weeks), wait for the viral expression (1 month), and perform the electrophysiological recordings (2 months). We agree that these experiments would indeed be interesting, but our laboratory has limited resources and we hope that the reviewers might agree that 18 months of revisions is an extremely long period of time, particularly because these experiments will not affect the main conclusion of the paper, which is that inhibiting the CRF^{CeA-BNST} projection decreases escalated alcohol drinking in dependent animals (note that we toned down the interpretation of the negative results and moved the negative results to the Supplementary Materials). Moreover, inhibition of the terminals through halorhodopsin has been successfully validated in previous studies (Seif, et al. 2013; Tye, et al. 2011) as well as in the current report.

Please note also that we added 5 control experiments in this revised version of the manuscript, including an experiment for each brain region where we showed that light *per se* (in animals that were injected with control virus) did not affect alcohol intake in dependent animals. This is important control because it rules out possible nonspecific effects of optogenetic stimulation, further confirming our main findings on the CRF^{CeA-BNST} projection. Finally, as mentioned by one of the reviewers, slice electrophysiology does not necessarily reflect *in vivo* conditions. Even if we performed these additional experiments and observed positive results, one could argue that it does not reflect physiological conditions. We again hope that the reviewers will agree that adding 9 months of additional work is unnecessary to confirm the negative results (which were not the main focus of the study). We believe that these new data with the control virus, together with the electrophysiological data that show a reduction of neuronal activity in the BNST after the light inhibition of CRF^{CeA-BNST} terminals, represent a proper validation of our model and solid controls for our main findings.

2) Related to the comment above, the most striking result of this study is that light activated HaloR in the CeA to BNST terminals (unilaterally) is enough to completely inhibit withdrawal-induced escalated EtOH consumption (Fig. 5B), despite having no impact on three of the 5 withdrawal symptoms tested (Fig. 5C; and unknown impact on other withdrawal symptoms). This raises the question of how such inhibition is causing such an outcome? The way the manuscript is introduced, one might infer that the mechanism is envisioned to be reduced withdrawal symptoms, and thus less drive for EtOH consumption as a negative reinforcer. However, this does not seem to fit with the fact that many withdrawal symptoms that are presumably aversive persist (Fig. 5C). An alternative possibility that is not considered is that the circuit plays a role in associative learning, in a manner that inhibition of the circuit inhibits memory of the ameliorative aspects of EtOH, which might then be expected to eliminate escalated EtOH consumption despite persistence of various aversive symptoms. It would increase the impact of this manuscript a great deal if the authors could discriminate between these two possibilities. Another possibility is that EtOH itself only reduces some of the aversive withdrawal symptoms, most parsimoniously the ones that are eliminated when CeA to BNST afferents are inhibited (abnormal gait & body tremors), and so optogenetic inhibition effectively

provides the same relief as EtOH, both being incomplete but equally reinforcing. It would be informative to determine what EtOH does to the tested withdrawal symptoms.

Response: These are interesting points. We incorporated them into the revised Discussion to read, “The present results also showed that the inhibition of CeA terminals in the BNST completely inhibited the withdrawal-induced escalation of alcohol consumption in dependent rats. The mechanism for this marked reduction of alcohol drinking might be related to a decrease in withdrawal symptoms that leads to less negative reinforcement associated with alcohol consumption. However, inhibition had no impact on three of the five withdrawal signs that were evaluated. One explanation for this result may be that somatic signs provide behavioral evidence of alcohol withdrawal, but they do not directly measure the negative affective state that is thought to drive the escalation of drinking. Another possibility may be that the CRF^{CeA-BNST} pathway plays a role in associative learning, and the inhibition of this pathway inhibits memories of the ameliorative effects of alcohol. Such a mechanism could be involved in mitigating the escalation of alcohol consumption despite the persistence of some withdrawal signs. Further studies are needed to discriminate between these different possibilities.”

3) A final concern, related to comments 1&2 above, is the fact that unilateral inhibition of the Crh CeA neurons or their afferents to the BNST completely prevents escalated EtOH consumption (Fig. 4B). This is somewhat surprising. Specifically, it is difficult to understand this complete block of escalated consumption in the context of the overall proposed model, i.e. that CIE-induced altered Crh-CeA signaling underlies the aversive aspects of withdrawal that drive escalated EtOH consumption as a negative reinforcer. Do the authors propose that one hemisphere of such alterations has no aversive impact? Perhaps relatedly, in Fig. 4H it appears that the image of the CeA is a mirror image orientation of the images in Fig4F&G (based on the position of the CeA relative to the direction of the tip of the nearby fissure). Is this the contralateral hemisphere? Either way it would be informative to examine C-Fos expression in the contralateral amygdala. Does unilateral inhibition somehow inhibit both hemispheres? If not, the authors should at least discuss this issue.

Response: We addressed this important point by providing an analysis of Fos expression in the contralateral CeA during optogenetic inhibition. The results showed that unilateral inhibition also affected the contralateral CeA. We observed a significant decrease in Fos⁺ neurons on the contralateral side. These results suggest that the reviewer is correct and that the CeA may be functionally connected bilaterally so that if one side is inhibited, then the other side is also inhibited. These data are now presented in Fig. 4.

4) The authors quantify EtOH consumption by “number of rewards” (presumably the number of active lever presses) during Operant self-administration trials. While this is an understandably easily quantified measurement, it does not inform us of the blood EtOH concentrations (BECs) achieved during such trials, which has several important ramifications and/or drawbacks. First, it does not seem appropriate to refer to such measurements as “excessive drinking”, page 4, 3rd sentence. All we can say is that drinking increased. Second, and more importantly, in the context of the EtOH consumption and escalation during EtOH withdrawal, it is implied that the driving force for elevated consumption is amelioration of the associated aversive state. Thus it is important to determine whether BECs achieved are actually high enough to significantly alter

CeA signaling. Such information might partially inform issue #2 above. Additionally, it is important to know what potential molecular mediators of such action are, which requires knowing the BECs achieved. I.e. are the levels achieved likely able to suppress CeA NMDARs, enhance GABAARs, alter transmitter release, alter Crh, etc. Thus, for both aspects, it is important to determine and present BECs achieved during baseline and elevated EtOH consumption.

Response: We now express the drinking data as g/kg of alcohol. Our dependent animals drank >1 g/kg per 30 min session. These amounts of alcohol have been shown to produce BALs of ~0.2 g%, which is known to produce multiple adaptations to the stress regulatory system, compromised hormonal responses, and sensitized brain stress responses in dependent animals (Richardson, et al. 2008).

5) There are a variety of issues that make it difficult to assess the outcomes of the immunocytochemical analysis and the relationship between the raw data and mean data as presented in the bar graphs (Figs 1E&F and Suppl. Fig. 1 C&D, and Fig. 2B). In particular, there is a considerable amount of ambiguity about the nature and magnitude of the staining to the naked eye in the raw data images, and there is an inadequate description of the quantitative details of how bar graphs were generated from such ambiguous images. There is also a lack of controls showing staining is specific and/or represents what the authors express in the bar graphs. Of specific concern in Fig. 1E&F is how the authors determined what counted as a c-Fos positive or negative cell. First, proper immunocytochemical controls showing that staining is specific to c-Fos are needed, or at least a citation to prior work showing that the specific antibody used is specific. Second, while the arrows appear to point to what the authors conclude are c-Fos positive cells (note, although deducible, the legend lacks a description of what the arrows and their color signify), there is clearly abundant green cytoplasmic staining in a much larger proportion of the exhibited cells but with a different apparent morphology (apparent cytoplasmic rings versus solid circles of color). What are the respective staining patterns? Is this an artifact of confocality (with rings versus solid circles depending on depth into a given cell)? Is the antibody specific? If the antibody is specific, what are the apparent cytoplasmic rings, and do they change (from the exemplar images it looks like rings disappear in withdrawal animals, does this reflect redistribution of c-Fos?), and what does this mean? Similarly, in Supplemental Fig. 1 C&D, while one can get a vague sense that arrows point to “darker” spots, it is not by any means clear. Basically every nucleus has some degree of brownish/black, and there is no description of how thresholding was determined or implemented, or how the accuracy of such implementation was confirmed? Furthermore, the dark spots do not appear to co-localize with the blue stain (presumed NeuN, indicative of neuronal subtype), which is supposed to indicate neuronal overlap. Again, a better description of methodology and confirmation of validity is needed.

Response: We thank the reviewer for these detailed comments. We clarified the immunohistochemistry methods by providing additional details on the validity of the antibody and analyses. We also provided additional images to address these concerns.

(1) The antibody that was used to detect Fos (Cell Signaling Technology, catalog no. 2250) has been used in over 71 publications as a specific marker for Fos (<https://www.citeab.com/antibodies/123097-2250-c-fos-9f6-rabbit-mab/publications>), as well as in our previous publications (de Guglielmo, et al. 2016; Leao, et al. 2015).

(2) We agree that there was a lot of background in the previous images. We now provide better representative photographs that are clearer. We classified neurons as Fos+/CRF+ when a neuron exhibited a focused red nucleus (Fos+) and a cytoplasm that was stained for CRF (green). Yellow arrows indicate Fos-/CRF+ cells. White arrows indicate Fos+/CRF+. Orange arrows indicate Fos+/CRF-. Three sections from each rat were bilaterally quantified within the boundary of the CeA and normalized by area, measured using Fiji software. Counts from all images for each rat were averaged so that each rat was an n of 1. The immunohistochemistry section in the Methods was revised to include this information.

(3) For DAB staining, brown (NeuN) and black (Fos) channels were separated using Fiji software to isolate the red channel. Using the red channel as the background channel for thresholding leaves the dark nuclei from the black (Fos) channel easy to identify and threshold. Thresholding was applied under the same conditions for both naive and alcohol-withdrawn rats. Thresholded images were counted for positive foci within the nucleus of NeuN-positive neurons. The immunohistochemistry section in the Methods was revised to include this information, and we added a sample image to Supplementary Fig. S1 (Fig. S1C).

6) There are some important issues with the electrophysiological analysis of the impact of HaloR activation on CeA neuronal excitability. First, the language describing the studies is confusing. The recordings are clearly in the current clamp configuration (i.e. clamping electrode current to record membrane potential), but the authors refer to holding potential (last sentences on page 5) which implies voltage-clamp (i.e. clamping membrane potential to record currents). This language needs to be fixed.

Response: We apologize for this mistake. The reviewer is correct that the experiment was performed in current-clamp mode. We rephrased this section of the Results and removed “held at” or “holding potential” to avoid confusion.

More importantly, there is no description of what the resting membrane potential was (and whether it was similar across conditions), or how much current was injected to alter the membrane potential to what one presumes was designed to be similar across cells?

Response: We now indicate the average resting membrane potential for the two groups. There was no significant difference between the two groups. We also added the average input resistance. There was no significant difference between the two groups; hence, similar current injections were needed to obtain similar depolarizations.

Most importantly though, this being a crucial control experiment, as best I can tell the authors have injected current to adjust the membrane potential to be right above action potential (AP) threshold, hence the steady action potential firing. While this approach provides some information (i.e. it clearly shows that light excitation of HaloR slightly hyperpolarizes CeA cells, and if they are just above AP threshold, it can silence their AP firing), it does not tell us much about the likely impact of light-activation of HaloR under normal physiological conditions, when cells are not manipulated, by constant current injection, to be just above AP threshold. What is more relevant is what impact such a small hyperpolarization has on post synaptic AP responses to signals from other cells, either in the form of glutamatergic/GABAergic or CRF transmission. This is particularly important given the very small hyperpolarization (5mV) induced by light

activation of HaloR. The authors should determine whether light excitation of HaloR can reduce AP output induced by afferent stimulation or CRF application, when the cells are not artificially depolarized with current injection.

A simpler but less elegant alternative, would be to examine the impact of light inhibition on responses to transient current injections from a cell at its native resting potential.

Response: This experiment was designed to confirm the validity of the optogenetic approach. As the reviewer pointed out, while having some drawbacks, the approach “clearly shows that light excitation of HaloR slightly hyperpolarizes CeA cells.” The spontaneous firing of CeA neurons at rest varies from cell to cell. Therefore, it is important to normalize these variations to be able to interpret the results without having to record an unreasonable number of cells. With intracellular recordings, spontaneous firing at rest can vary considerably from cell to cell. By depolarizing neurons, we ensured consistent and sustained firing to observe the influence of NpHR on firing rate. Additionally, we believe that a hyperpolarization of 6-7 mV qualifies as a strong effect rather than “very small,” although it is more likely to be 4-5 mV around the resting potential (still a sizeable effect). Being able to directly correlate the degree of hyperpolarization and decrease in spiking with behavioral observations would be exciting, but we feel that it would be quite a stretch. It must be kept in mind that we are using an *ex vivo* approach. Although brain slices can last several hours in excellent health, we remain far from “normal physiological conditions.” Consequently, one could argue that the resting potential of a neuron that is seen in the slice preparation may be different from the resting potential that is seen *in vivo*, and a non-spiking neuron in a slice could be a spiking neuron *in vivo*. Thus, pinpointing the effect of NpHR at resting potential to build an interpretation for the behavioral observations carries significant caveats. We agree that the alternative approaches that are proposed by the reviewer could provide additional information, but they would not change the original results and demonstration of the validity of the optogenetic approach in silencing CRF+ neurons that are firing.

Also, in both the existing experiment and any future experiments, the authors should provide detailed quantification of the change in AP output, as opposed to just stating that it “suppressed AP firing rate” (line 95).

Response: We now quantify the changes in action potential output.

I emphasize, combined with the behavioral data I don't doubt that the HaloR stimulation can reduce CeA firing *in vivo*, but it is important to know more realistically how such behavioral changes are induced. As the figure stands it might be concluded that HaloR stimulation completely shuts down CeA firing, which is highly unlikely to be the case under more physiological circumstances. This is important for our understanding of how much cellular inhibition is required to implement such dramatic behavioral outcomes.

Response: We fully agree with the reviewer that the all-or-none figure may lead the reader to conclude that HaloR shuts down CeA activity. We quantified the reduction of action potential firing by HaloR and now emphasize “reduced spiking activity” with HaloR. Additionally, our new results in the BNST show lower spiking activity instead of a shutdown (see Fig. 5).

7) The graphical display of withdrawal symptom subsets is too small (Figs. 4C and 5C-L). While I appreciate the value of the combined score insert, the potentially more informative comparisons of withdrawal subset comparisons across Crf-CeA inhibition and CeA to BNST afferent inhibition is very difficult to see given the scaling required to accommodate the insert. The Y-axis scale of the subset graphs should be expanded so that comparisons across the two conditions can be more easily made. The large size of combined totals insert, while informative, obscures differences in the subcategories. In this regard, it appears that the only subcategory that is significantly reduced by both optical inhibition of the Crf-CeA and CeA to BNST afferents is “abnormal gait”, despite both completely inhibiting escalated EtOH consumption. Does this imply abnormal gait is the main behavioral manifestation of EtOH withdrawal that drives escalation of EtOH consumption? If so, how does this relate to the known functions of CeA to BNST circuit, and/or downstream aspects of the circuit?

Response: As suggested by the reviewer, the Y-axis scale for the individual signs was expanded. However, the withdrawal signs are very variable between individuals, and this is the reason why we and other groups show the sum of the individual signs as a “Total Withdrawal Score.” For references, see (Braconi, et al. 2010; de Guglielmo, et al. 2016; de Guglielmo, et al. 2017b; de Guglielmo, et al. 2015; Macey, et al. 1996; Sidhpura, et al. 2010). It is important to clarify that we do not believe that there is necessarily a causal relationship between the reductions of some of the somatic signs and alcohol drinking. We clarified this point in the Discussion.

Minor comments:

1) The authors need to refer to a manuscript that describes the generation and characterization of the Crh-Cre rats used in this study. Apologies if I missed it somewhere, but it is not in the first reference to the animals or in the methods that describe their use as subjects.

Response: We cited the following article in the Introduction and Electrophysiology and Subjects sections of the Methods: Pomrenze (2015) A Transgenic Rat for Investigating the Anatomy and Function of Corticotrophin Releasing Factor Circuits. *Front Neurosci* 9, 487.

2) The authors should explicitly state the precise time frame (relative to CIE termination) for when operant self-administration and withdrawal symptoms were quantified.

Response: This information is in the Methods section (Alcohol vapor chambers):
“...the rats were made dependent by chronic intermittent exposure to alcohol vapors. They underwent cycles of 14 h ON (blood alcohol levels during vapor exposure ranged between 150 and 250 mg%) and 10 h OFF, during which behavioral testing for acute withdrawal occurred (i.e., 6-8 h after the vapor was turned OFF, when brain and blood alcohol levels are negligible).”

Reviewer 2

The manuscript by George and colleagues seeks to assess the role for neuronal activations of the central nucleus of the amygdala (CeA) aspects of alcohol behaviors that are associated with dependence/compulsive drinking in rodent models. The overall hypothesis was that corticotropin-releasing factor (CRF) neurons and downstream targets, such as their known

projections to the bed nucleus of the stria terminalis (BNST), might be involved and that these CRF neurons and related circuitry might be a potential target for reducing compulsive drinking in AUD. These authors have utilized a number of state-of-the-art techniques to test their hypothesis, and have provided a well written and compelling study. The hypothesis that the CeA is involved in AUD is solidly based on a large pre-clinical literature, notably that neuronal ensembles in the CeA are associated with alcohol withdrawal behaviors in alcohol-dependent rats. A large body of evidence implicates CRF neurotransmission in the CRF in withdrawal and negative emotional states, and supportive pharmacological evidence for a CRF CeA link. In this study, first Crh-Cre transgenic rats were used in an animal model of alcohol dependence combined with in vivo optogenetics and immediate early gene brain mapping. Then the role for several possible downstream CeA-CRF projections were assessed by optogenetic inactivation methods, i.e., CRF terminals from CeA neurons that project to the BNST (CRFCeA-BNST), LH and pSTN (CRFCeA-LH/pSTN), substantia innominata (SI; CRFCeA-SI), and parabrachial nucleus (PBN; 59 CRFCeA-PBN).

The main findings of these experiments were that 1) withdrawal from alcohol dependence significantly and mostly affects CRF neurons (~ 80%), 2) that the inactivation of these CeA CRF+ neurons prevents the recruitment of the CeA neuronal ensemble, and reduces dependence-associated alcohol drinking and withdrawal, and 3) that based on the optogenetic circuitry assessment that it is activation of CeA neurons that project to the BNST that are critically involved. The paper is exceptionally well prepared and generally presented.

Response: We thank the reviewer for the positive evaluation of the manuscript.

There are several issues of interpretation and some details that need to be addressed. The conclusion that CRF neurons are critical to “drive compulsive-like alcohol drinking” is unclear. Although dependent rats were found to have a significant proportion of the CeA neuronal ensemble that is associated with withdrawal in alcohol-dependent rats, it seems a stretch to argue that this reflects compulsive (or compulsive-like) behavior, or what component of “compulsion”. While these data do seem to support the overall conclusion that there are key CeA neuronal ensembles that are involved in alcohol drinking in dependent rats – nor is clear if this is only a result from the vapor-inhalation model. These data do nonetheless show that inactivation of the CRFCeA-BNST pathway – rather selectively -- could reverse this “excessive” alcohol drinking during withdrawal and partially prevented the somatic signs of withdrawal. More discussion and validation is needed to argue that these data have identified a cellular mechanism of excessive alcohol use, and it appears that with regards to this statement in the Introduction they cite their own JN paper. Indeed, in the Results the word “escalation” of intake is used to describe these actual data (Figure 1A), though then at other points it is then referred to as “excessive”, and then as noted above, the term “compulsive” then gets introduced when these -- though the title and one place in the Discussion uses the careful “addiction-like” term, even though the Discussion starts with “compulsive-like”. I think that words matter.

Response: We thank the reviewer for this comment. We now refer only to the escalation of alcohol intake.

The details with regards to numbers of animals used for the statistical analyses are not transparent in several places. In fact (perhaps I missed it) I cannot find any details with regards to

number of animals for each experiment/condition with the exception of the whole-cell recordings. Statistical analyses section is too general and does not make it clear the numbers of animals that were used/reported in these data presented in the figure legends.

Response: The number of rats used in each experiment is now stated in the text.

A number of minor issues need to be addressed.

It would be helpful that scale the axis for comparisons to be matched. For example in Figure 5 the axis showing the rewards with the optogenetic inhibition of the inhibition BNST is not the same as for the SI, LH, and PBN for the alcohol SA data.

Response: As suggested by Reviewer 3, we reported all of the alcohol data as grams of alcohol per kilogram of body weight. We also ensured that all scales are the same for axis comparisons.

There is no justification for the exclusive use on male rats, and given NIH guidelines female rats should be included.

Response: The integration of female data would be really interesting, but it is beyond the scope of the present study. This project started several years before NIH's new guidelines on female studies, and it would take several years to perform all of these experiments again in females. We added a sentence in the manuscript to address this limitation.

There are scant details with regards to the operant self-administration and methods used to “get” animals to drink the 10% (w/v) alcohol, and for the saccharin self-administration study, and whether the alcohol animals were initially trained to drink saccharin and alcohol in a “fading” design.

Response: The animals were not pretrained to lever press for saccharin and did not undergo a fading procedure. The self-administration training is very straightforward and is described in the methods “For the alcohol self-administration studies, the animals were first trained to self-administer 10% (v/v) alcohol and water solutions until a stable response pattern (20 ± 5 rewards) was maintained. The rats were subjected to an overnight session in the operant chambers with access to one lever (right lever) that delivered water (fixed-ratio 1 [FR1]). Food was available *ad libitum* during this training. After 1 day off, the rats were subjected to a 2 h session (FR1) for 1 day and a 1 h session (FR1) the next day, with one lever delivering alcohol (right lever). All of the subsequent alcohol self-administration sessions lasted 30 min. The rats were allowed to self-administer a 10% (v/v) alcohol solution (right lever) and water (left lever) on an FR1 schedule of reinforcement (i.e., each operant response was reinforced with 0.1 ml of the solution).”

These studies are based on a model of alcohol dependence produces by exposure to chronic intermittent alcohol vapor. While many groups use this model, it is not clear to the extent this model has clear translational relevance and that these effects can be ascribed selectively to alcohols pharmacological effects as opposed to more general effects of stress of aversion. For example, while the number of Fos+/CRF+ neurons in the CeA significantly increased during withdrawal this is based on the comparison between alcohol-withdrawn and alcohol-naive rats

and in my opinion the alcohol-naïve rats are not an appropriate control. Given the above concern, the authors could use other alcohol drinking models other than vapor inhalation.

Response: The model of chronic intermittent exposure (CIE) to alcohol vapor has been shown to have robust predictive validity for alcoholism and construct validity for the neurobiological mechanisms of alcohol dependence (Heilig and Koob 2007; Koob 2009). Indeed, rats that are made dependent by CIE exhibit clinically relevant blood alcohol levels (BALs; 150-250 mg/100 ml), an increase in alcohol drinking when tested during early and protracted abstinence, and importantly compulsive-like alcohol drinking (e.g., responding despite adverse consequences; (Kimbrough, et al. 2017b; Leao, et al. 2015; O'Dell, et al. 2004; Roberts, et al. 1996; Schulteis, et al. 1995; Vendruscolo, et al. 2012). Alcohol dependence that is induced by alcohol vapor also results in withdrawal symptoms during both acute withdrawal and protracted abstinence (de Guglielmo, et al. 2017a; Kallupi, et al. 2014; Macey, et al. 1996; Vendruscolo and Roberts 2014), anxiety-like behavior (Valdez, et al. 2002), irritability-like behavior (Kimbrough, et al. 2017a), and the development of mechanical hyperalgesia (Edwards, et al. 2012). With regard to the increase in Fos+/CRF+ neurons during withdrawal in dependent rats, we understand that the reviewer feels that a nondependent control would have been more appropriate. However, it has been shown that withdrawal from alcohol in nondependent subjects does not induce Fos activation in the CeA (George, et al. 2012; Weitemier, et al. 2001). In the present study (Fig. 3A), we found that the inactivation of CeA CRF neurons did not affect alcohol intake in nondependent rats.

These whole-cell patch electrophysiological data to confirm the efficacy of NpHR in preventing CRF+ neuron firing in acute CeA slices from Crh-Cre rats was not well described or integrated into the manuscript.

Response: The electrophysiology section has been reorganized and improved to provide additional details and data (see also response to Reviewer 1).

It is confusing that the Figures with legends are presented before the References cited, and then shown again without the legends at the end of the manuscript.

Response: The manuscript has been reorganized, and the figure legends are presented only once.

Reviewer 3

Major Concerns:

1) The authors examine many potential outputs of CeA CRF neurons including the Bed Nucleus of the Stria Terminalis (BNST), Substantia Inominata (SI), Lateral Hypothalamus (LH), and Parabrachial Nucleus (PBN). While some of these outputs are well-known, such as the BNST and PBN, others, such as the SI and LH, have not been described previously. Given light microscopy cannot differentiate between terminals and fibers of passage, it is unclear whether the lack of effects seen in these regions are true effects or are simply without effect due to the lack of synapses present. The authors must test the function of these outputs in order to claim that inhibition of these outputs are without effect. One potential way to resolve this question

would be to perform light-evoked recordings in this region and examine what percentage of cells are responsive to stimulation.

2) Going further, what is the overlap between BNST-projecting CRF neurons and the other outputs tested? Are BNST projections are separate population or are they collaterals of the same cells?

Response: These are very interesting points. We agree that the negative results with the other pathways could potentially be attributable to either fiber of passage or technical limitations, or these pathways may play roles in other withdrawal-related behaviors. However, testing these novel hypotheses would require several years of work and are beyond the scope of the present study. Moreover, whether the BNST projections are separate populations or collaterals of the same cells would not affect the main conclusion of the study, which is that CeA-to-BNST CRF neurons are required for excessive alcohol drinking and withdrawal symptoms in dependent rats. We toned down the negative results to emphasize the positive results that were obtained with the BNST, for which we performed an additional experiment to test the effect of inhibition of the terminals. We moved the negative results from the main section of the manuscript to the Supplementary Material to ensure transparency and because we believe that this information may still be very useful to other laboratories that work on these pathways.

3) The authors find that both somatic inhibition and inhibition of BNST outputs produce similar effects. This does not preclude a role for local CRF neurons in the CeA in regulating alcohol intake. Given the body of work describing the effects of alcohol on CRFR1+ cells in the CeA (Herman et al. 2013; Herman et al. 2016), it would be helpful to examine a role for CRF neurons in the CeA using an intersectional strategy to target non-BNST projecting neurons. A major limitation of this paper is lack of any evidence describing a role for CRF signaling itself. The authors acknowledge that CeA CRF neurons also express a medley of other neuropeptides such as dynorphin, neotensin, and somatostatin, all of which may be contributing to the effects seen. Thus it is unclear the whether the results seen are due to manipulation of CRF signaling or inhibition of a unique subset of CeA neurons.

Response: The hypothesis of the study was not to demonstrate that CRF peptide is the sole contributor to alcohol drinking. We fully agree with the reviewer that other neurotransmitters and neuromodulators very likely contribute to the observed effects. We tested the role of CRF expressing neurons and their pathways in alcohol drinking. Dissecting the role of the different peptides and neurotransmitters within these neurons is a very interesting but monumental task and beyond the scope of the present study.

Another major limitation of this study is the lack of appropriate non-dependent controls. Studies using CIE should include air exposed controls to show that effects are due induction of alcohol-dependence. More seriously, there are no virus controls for the experiments in figure 5. Without these data points, it is not possible to determine whether the experiments in this aim are the result are a true effect of inhibition of the pathway or a spurious result of laser stimulation.

Response: The effect of inhibiting CeA CRF neurons in nondependent subjects is described in Fig. 3A. We showed that halorhodopsin inhibition of CRF neurons in the CeA did not affect alcohol self-administration in nondependent rats. We also added no-virus controls for each CeA

terminal output that was tested. We found that light activation of each of the projection sites did not affect alcohol self-administration in dependent rats.

The author claim they targeted CRF neurons in the CeL, however there is also a population of CRF neurons in the CeM. Given the lack of clear anatomical boundaries between these two subdivisions, it is difficult to believe that expression was constrained to the CeL.

Response: We agree with the reviewer and now use the term CeA throughout the paper instead of CeL.

Throughout the manuscript statistics are poorly described, and in several cases do not match the data shown.

Response: We carefully checked and updated all of the statistics and improved the descriptions of the results. The statistical analysis paragraph in the methods section now reads: “The data are expressed as mean \pm SEM. For comparisons between only two groups, p values were calculated using paired or unpaired t -tests as described in the figure legends. Comparisons across more than two groups were made using one-way analysis of variance (ANOVA), and two-way ANOVA was used when there was more than one independent variable. Experiments that tested the effect of the unilateral optogenetic inhibition of CeA CRF neurons on the escalation of alcohol self-administration in dependent rats were analyzed using a two-way repeated-measures ANOVA, with time and laser illumination as within-subjects factors (see Fig. 4, 5, S4, S5, and S6). In the same experiments, differences between baseline intake before dependence and intake after vapor exposure (ON and OFF days; see Fig. 4, 5, S4, S5, and S6) were analyzed using separate one-way repeated-measures ANOVAs. The Newman Keuls *post hoc* test was used following significance in the ANOVA. Withdrawal signs were analyzed using the nonparametric Mann-Whitney U statistic, followed by Dunn’s multiple-comparison test. The standard error of the mean is indicated by error bars for each group of data. Differences were considered significant at $p < 0.05$. All of the data were analyzed using Statistica 7 software.

The authors do not demonstrate effective presynaptic inhibition using the in vivo parameters. This is a topic of great debate and needs to be included.

Response: This is a very important point, and we performed slice experiments to address this issue. The results are presented in Fig. 5, showing that the inhibition of axon terminals in the BNST is sufficient to reduce neuronal activity, confirming the behavioral data.

Minor Concerns:

For introduction cite original research papers rather than reviews?

Response: Citations from original research papers have been added to the Introduction.

What quantity of alcohol is administered by the rats? All the figures show the number of rewards obtained, however it would be more informative to report this as grams of ethanol per kilogram of body weight. This allows the readers to understand the effective dose of alcohol animals are receiving and would normalize differences in body weight across animals.

Response: All of the alcohol data are now expressed as grams of alcohol per kilogram of body weight, as suggested by the reviewer.

The Whole-Cell Patch Clamp recording section begins rather abruptly. The manuscript would read more smoothly if a few sentences of introduction were added stating that the purpose of these experiments were to validate the efficacy of DIO-NpHR in the CeA of CRF-Cre rats. Likewise, it would be helpful to have a more descriptive title such as “Halorhodopsin effectively inhibits CeA CRF neurons” rather than just calling it “Whole-Cell Patch Clamp Recording”

Response: The electrophysiological validation of the efficacy of NpHR in preventing CRF+ neuron firing section has been modified accordingly to the reviewer’s suggestion.

In the methods section, a specific age range of animals should be given rather than listing them as “adults”

Response: Done.

References

- Braconi, S., et al.
2010 Revisiting intragastric ethanol intubation as a dependence induction method for studies of ethanol reward and motivation in rats. *Alcohol Clin Exp Res* 34(3):538-44.
- de Guglielmo, G., et al.
2016 Recruitment of a Neuronal Ensemble in the Central Nucleus of the Amygdala Is Required for Alcohol Dependence. *J Neurosci* 36(36):9446-53.
- de Guglielmo, G., et al.
2017a Voluntary induction and maintenance of alcohol dependence in rats using alcohol vapor self-administration. *Psychopharmacology (Berl)* 234(13):2009-2018.
- 2017b Voluntary induction and maintenance of alcohol dependence in rats using alcohol vapor self-administration. *Psychopharmacology (Berl)*.
- de Guglielmo, G., et al.
2015 MT-7716, a potent NOP receptor agonist, preferentially reduces ethanol seeking and reinforcement in post-dependent rats. *Addict Biol* 20(4):643-51.
- Edwards, S., et al.
2012 Development of mechanical hypersensitivity in rats during heroin and ethanol dependence: alleviation by CRF(1) receptor antagonism. *Neuropharmacology* 62(2):1142-51.
- George, O., et al.
2012 Recruitment of medial prefrontal cortex neurons during alcohol withdrawal predicts cognitive impairment and excessive alcohol drinking. *Proc Natl Acad Sci U S A* 109(44):18156-61.
- Heilig, M., and G. F. Koob
2007 A key role for corticotropin-releasing factor in alcohol dependence. *Trends Neurosci* 30(8):399-406.

- Kallupi, M., et al.
2014 Neuropeptide YY(2)R blockade in the central amygdala reduces anxiety-like behavior but not alcohol drinking in alcohol-dependent rats. *Addict Biol* 19(5):755-7.
- Kimbrough, A., et al.
2017a CRF1 Receptor-Dependent Increases in Irritability-Like Behavior During Abstinence from Chronic Intermittent Ethanol Vapor Exposure. *Alcohol Clin Exp Res*.
- Kimbrough, A., et al.
2017b Intermittent Access to Ethanol Drinking Facilitates the Transition to Excessive Drinking After Chronic Intermittent Ethanol Vapor Exposure. *Alcohol Clin Exp Res* 41(8):1502-1509.
- Koob, G. F.
2009 Brain stress systems in the amygdala and addiction. *Brain Res* 1293:61-75.
- Leao, R. M., et al.
2015 Chronic nicotine activates stress/reward-related brain regions and facilitates the transition to compulsive alcohol drinking. *J Neurosci* 35(15):6241-53.
- Macey, D. J., et al.
1996 Time-dependent quantifiable withdrawal from ethanol in the rat: effect of method of dependence induction. *Alcohol* 13(2):163-70.
- O'Dell, L. E., et al.
2004 Enhanced alcohol self-administration after intermittent versus continuous alcohol vapor exposure. *Alcohol Clin Exp Res* 28(11):1676-82.
- Richardson, H. N., et al.
2008 Alcohol self-administration acutely stimulates the hypothalamic-pituitary-adrenal axis, but alcohol dependence leads to a dampened neuroendocrine state. *Eur J Neurosci* 28(8):1641-53.
- Roberts, A. J., M. Cole, and G. F. Koob
1996 Intra-amygdala muscimol decreases operant ethanol self-administration in dependent rats. *Alcohol Clin Exp Res* 20(7):1289-98.
- Schulteis, G., et al.
1995 Decreased brain reward produced by ethanol withdrawal. *Proc Natl Acad Sci U S A* 92(13):5880-4.
- Seif, T., et al.
2013 Cortical activation of accumbens hyperpolarization-active NMDARs mediates aversion-resistant alcohol intake. *Nat Neurosci* 16(8):1094-100.
- Sidhpura, N., F. Weiss, and R. Martin-Fardon
2010 Effects of the mGlu2/3 agonist LY379268 and the mGlu5 antagonist MTEP on ethanol seeking and reinforcement are differentially altered in rats with a history of ethanol dependence. *Biol Psychiatry* 67(9):804-11.
- Tye, K. M., et al.
2011 Amygdala circuitry mediating reversible and bidirectional control of anxiety. *Nature* 471(7338):358-62.
- Valdez, G. R., et al.
2002 Increased ethanol self-administration and anxiety-like behavior during acute ethanol withdrawal and protracted abstinence: regulation by corticotropin-releasing factor. *Alcohol Clin Exp Res* 26(10):1494-501.
- Vendruscolo, L. F., et al.

- 2012 Corticosteroid-dependent plasticity mediates compulsive alcohol drinking in rats.
J Neurosci 32(22):7563-71.
- Vendruscolo, L. F., and A. J. Roberts
2014 Operant alcohol self-administration in dependent rats: focus on the vapor model.
Alcohol 48(3):277-86.
- Weitemier, A. Z., et al.
2001 Expression of c-Fos in Alko alcohol rats responding for ethanol in an operant
paradigm. Alcohol Clin Exp Res 25(5):704-10.

Reviewers' Comments:

Reviewer #1:

Remarks to the Author:

This manuscript is a resubmitted version after revision. The background, significance of the topic, overall experimental design, results and conclusions have not changed substantially (see original review for details).

The authors have made a sincere effort to address my previous concerns, including conducting significant new experimental controls, additional quantification and improved data presentation, with relevant associated revised text. Importantly, new controls showing optical inhibition of Arch-expressing synaptic terminals effectively reduces excitation of neurons in BNST slices significantly increases confidence that optical inhibition of such terminals in vivo does in fact underlie the observed behavioral outcomes. Where the authors chose not to execute suggested new experiments, they provide reasonable argument against the practicality of doing so (largely unreasonable time required given the relevance of the specific experiments avoided), and have adapted the text appropriately to focus on the more controlled and more central experiments, and acknowledging the resultant limitations. The original manuscript was already quite solid and interesting, and the revisions have made it even more so. All around a very high quality and intriguing study about a very critical biomedical condition, alcohol use disorder.

Minor comment:

Unless I am misunderstanding the affiliation formatting, the authors seem to have swapped author affiliation addresses for Drs. Koob and Messing.

Reviewer #2:

Remarks to the Author:

The authors have addressed all my comments, and seem to have addressed many related comments from the other reviewers. The manuscript is much improved.

My only comment is that these authors change "the data" to "these data" in the manuscript.

Reviewer #3:

Remarks to the Author:

The authors have performed additional experiments to address the issues raised in the last round of review. While several of the previous concerns were addressed, several issues remain.

1) The authors demonstrate that a brief photoinhibition of CRF inputs from the CeA to the BNST can reduce firing of BNST neurons. The authors suggest that this is due to an inhibition of CRF release. There are multiple issues raised here. This is a very important issue, as it speaks to the fundamental idea of the manuscript, that some interaction between the CeA and BNST drives alcohol consumption following escalation, consistent with several pieces of data in the field. However, this data as presented is problematic as it still does not inform the nature of the connection between these regions, crucial for understanding and interpreting optogenetic studies.

a) The authors suggest that a brief photoinhibition will reduce CRF release, reducing CRFR1 activation which potentiates glutamate release, which leads to reduced firing. This is a testable mechanism, and

gets to the core of the matter of their findings. The authors in their reply to the reviewers state that "The hypothesis of the study was not to demonstrate that CRF peptide is the sole contributor to alcohol drinking. We fully agree with the reviewer that other neurotransmitters and neuromodulators very likely contribute to the observed effects." We are not asking that they look at other neurotransmitters and neuromodulators, but rather more precisely establish a role for CRF signaling itself, as their own data suggests it drives changes in connectivity. For example in figure 5E the authors show that inhibition of CeA projections to BNST result in reductions in firing in BNST neurons. Given that they are inhibiting a GABAergic projection, it is strange that they find a decrease firing rather than increase. The authors claim that this may be due to CRF potentiation of glutamate release, however this unlikely to occur on this timescale. Previously it has been shown that CRF increases glutamate release on the timescale of 5-10 min (Kash, Nobis, et al 2008). Thus it is unlikely that the authors find fast-acting reversible changes within 2 seconds of stimulation. If the authors want to make the claim that this is a CRF-dependent process they should perform these recordings with pretreatment of a CRFR1 antagonist. Otherwise they should more closely assess what fast-acting neurotransmitter is mediating the effect and what effect this has on BNST neurons. For example, it is well within the realm of possibility that the CeA CRF neurons inhibit a population of BNST GABA neurons that drive feedforward inhibition, leading to excitation. I am not suggesting doing all of this in vivo, but you have a preparation which can allow for testing of the core aspects of this mechanisms.

b) the in vivo stimulation paradigm used was either 5 minutes of constant stimulation or 30 minutes of constant stimulation. Both of these stimulations can produce heating (<https://www.ncbi.nlm.nih.gov/pmc/articles/PMC4512881/>) which can impact neuronal function. It is unclear why the authors did not use their in vivo stimulation paradigms for the slice work. Again, this impacts the interpretations of the in vivo studies. If 5 minutes of green light drives an alteration of connectivity, then this would inform the greater hypothesis. In addition, it is unclear why the authors did not provide any converging mechanisms for this finding, which would have greatly strengthened the conclusions.

c) Minor point: if the authors are going to select out a population that shows an inhibition, then they should show the 'other' population as well so readers can understand the spread between groups. Also, it is unclear as to the cutoff that the authors used to denote inhibition of firing.

2) The authors provide data demonstrating that in vivo inhibition of the CeA CRF pathway to the BNST does not impact baseline, non-escalated, drinking. Can they please clarify the time point, post injection when this was done. It is unclear if the lack of an effect was due to lack of expression at opsin at the terminal, as compared to the post escalation which was done at a much later time point. For all of the experiments, timelines would aid in interpretation and understanding.

3) The inclusion of the additional regions as negative findings, without any idea as if they were validated is problematic. I understand the author's position that this represents a great deal of work, but without further validation, these results are very difficult to appropriately interpret. This important caveat should be emphasized.

4) The authors demonstrate on-off effects for Halorhodopsin animals are reversible across 6 test days, however they do not show this for control animals. The control animals should be plotted on the same graph as the Halorhodopsin animals is should be shown as it will allow readers to more clearly understand the null findings.

5) The authors state "We also added no-virus controls for each CeA terminal output that was tested." However in the figures they are listed as DIO-eYFP controls, please clarify. Performing light inhibition in the absence of a viral control is very unusual and the authors should provide a justification for doing

so. Additionally, as these controls did not appear in the initial manuscript raises the question of whether they were run a later date. If so then this should be explicitly acknowledged in the manuscript. In addition, it is unclear if all of the other outputs were assessed at low drinking levels, as such this is an important caveat that should be included in the discussion.

Minor concerns:

1. Figure 2A lacks any anatomical landmarks showing the extent of viral expression inside and outside of the CeA. In addition, the photomicrographs in figures 2, 4, 5, and the supplemental figures are all missing scale bars.

Response to Reviewers

Reviewer 1

This manuscript is a resubmitted version after revision. The background, significance of the topic, overall experimental design, results and conclusions have not changed substantially (see original review for details).

The authors have made a sincere effort to address my previous concerns, including conducting significant new experimental controls, additional quantification and improved data presentation, with relevant associated revised text. Importantly, new controls showing optical inhibition of Arch-expressing synaptic terminals effectively reduces excitation of neurons in BNST slices significantly increases confidence that optical inhibition of such terminals in vivo does in fact underlie the observed behavioral outcomes. Where the authors chose not to execute suggested new experiments, they provide reasonable argument against the practicality of doing so (largely unreasonable time required given the relevance of the specific experiments avoided) and have adapted the text appropriately to focus on the more controlled and more central experiments and acknowledging the resultant limitations. The original manuscript was already quite solid and interesting, and the revisions have made it even more so. All around a very high quality and intriguing study about a very critical biomedical condition, alcohol use disorder.

Response: We thank the reviewer for the positive evaluation of the manuscript.

Minor comment:

Unless I am misunderstanding the affiliation formatting, the authors seem to have swapped author affiliation addresses for Drs. Koob and Messing.

Response: The reviewer is correct. This has been fixed.

Reviewer 2

The authors have addressed all my comments and seem to have addressed many related comments from the other reviewers. The manuscript is much improved.

Response: We thank the reviewer for the positive evaluation of the manuscript

My only comment is that these authors change "the data" to "these data" in the manuscript.

Response: We made the changes as suggested by the reviewer.

Reviewer 3

The authors have performed additional experiments to address the issues raised in the last round of review. While several of the previous concerns were addressed, several issues remain.

1) The authors demonstrate that a brief photoinhibition of CRF inputs from the CeA to the BNST can reduce firing of BNST neurons. The authors suggest that this is due to an inhibition of CRF release. There are multiple issues raised here. This is a very important issue, as it speaks to the fundamental idea of the manuscript, that some interaction between the CeA and BNST drives alcohol consumption following escalation, consistent with several pieces of data in the field. However, this data as presented is problematic as it still does not inform the nature of the connection between these regions, crucial for understanding and interpreting optogenetic studies.

Response: We thank the reviewer for providing very constructive comments. We performed the additional experiments as suggested by the reviewer and added new data to the manuscript (described below). We agree with the reviewer that these new experiments considerably strengthen the manuscript. (see responses to comments below)

a) The authors suggest that a brief photoinhibition will reduce CRF release, reducing CRFR1 activation which potentiates glutamate release, which leads to reduced firing. This is a testable mechanism and gets to the core of the matter of their findings. The authors in their reply to the reviewers state that “The hypothesis of the study was not to demonstrate that CRF peptide is the sole contributor to alcohol drinking. We fully agree with the reviewer that other neurotransmitters and neuromodulators very likely contribute to the observed effects.” We are not asking that they look at other neurotransmitters and neuromodulators, but rather more precisely establish a role for CRF signaling itself, as their own data suggests it drives changes in connectivity. For example, in figure 5E the authors show that inhibition of CeA projections to BNST result in reductions in firing in BNST neurons. Given that they are inhibiting a GABAergic projection, it is strange that they find a decrease firing rather than increase. The authors claim that this may be due to CRF potentiation of glutamate release, however this unlikely to occur on this timescale. Previously it has been shown that CRF increases glutamate release on the timescale of 5-10 min (Kash, Nobis, et al 2008). Thus, it is unlikely that the authors find fast-acting reversible changes within 2 seconds of stimulation. If the authors want to make the claim that this is a CRF-dependent process they should perform these recordings with pretreatment of a CRFR1 antagonist. Otherwise they should more closely assess what fast-acting neurotransmitter is mediating the effect and what effect this has on BNST neurons. For example, it is well within the realm of possibility that the CeA CRF neurons inhibit a population of BNST GABA neurons that drive feedforward inhibition, leading to excitation. I am not suggesting doing all of this in vivo, but you have a preparation which can allow for testing of the core aspects of this mechanism.

Response: We thank the reviewer for this important comment. We followed the reviewer’s suggestion and performed cell recordings in the BNST after pretreatment with the CRF₁ receptor antagonist R121919 and in combination with optogenetic inhibition. These results are presented in Fig. 5H and 5I, showing that pretreatment with R121919 mimicked the effect of optogenetic inhibition, leading to a reduction of cell firing, and pretreatment with R121919 occluded the effect of optogenetic inhibition, suggesting that the effect of optogenetic inhibition is mediated by the activation of CRF₁ receptors. Regarding the timeline of response, we found that ~50 percent of the cells showed a response when the optogenetic inhibition was performed for only 6.5 seconds, but all the cells showed reduced firing when the inhibition lasted 5 minutes (Fig 5, S4 and S8). In fact, for the BNST cells, the average latency of response to the laser was ~30 sec

with a maximum effect after 3 minutes. In the R121919 experiment the latency for response was ~3 minutes (which is close to the time required for the perfusion of the slice). This is in line with previous results that show that CRF can produce calcium release in less than 5 seconds in the pituitary (Watanabe and Orth 1987, Deng, Riquelme et al. 2015) and with what is shown in the paper cited by the reviewer (Kash, Nobis et al. 2008) where in fig 5A CRF enhances glutamate release in the BNST 2-4 minutes after the bath application on the slices.

b) the in vivo stimulation paradigm used was either 5 minutes of constant stimulation or 30 minutes of constant stimulation. Both of these stimulations can produce heating (<https://www.ncbi.nlm.nih.gov/pmc/articles/PMC4512881/>) which can impact neuronal function. It is unclear why the authors did not use their in vivo stimulation paradigms for the slice work. Again, this impacts the interpretations of the in vivo studies. If 5 minutes of green light drives an alteration of connectivity, then this would inform the greater hypothesis. In addition, it is unclear why the authors did not provide any converging mechanisms for this finding, which would have greatly strengthened the conclusions.

Response: This is another important point raised by the reviewer. We performed additional experiments to test the effect of 5 min of inhibition on cell firing in the CeA and BNST. The results show that 5 min of optical inhibition in the CeA and BNST reduced cell firing in both regions (Supplementary Fig. S4).

c) Minor point: if the authors are going to select out a population that shows an inhibition, then they should show the 'other' population as well, so readers can understand the spread between groups. Also, it is unclear as to the cutoff that the authors used to denote inhibition of firing.

Response: We now provide these data in Supplementary Fig. S8. We selected the two populations based on their response to light inhibition. Cells that showed more than a 10% reduction of firing rate during light inhibition were considered responders. This is now clarified in the text.

2) The authors provide data demonstrating that in vivo inhibition of the CeA CRF pathway to the BNST does not impact baseline, non-escalated, drinking. Can they please clarify the time point, post injection when this was done? It is unclear if the lack of an effect was due to lack of expression at opsin at the terminal, as compared to the post escalation which was done at a much later time point. For all of the experiments, timelines would aid in interpretation and understanding.

Response: Timelines for the experiments are now provided in Fig. 3, 4, 5, S5, S6, and S7. The experiment that tested the effect of optogenetic inhibition on non-escalated drinking was performed 5 weeks after the virus injection (Fig. 3B). Based on the literature, it is extremely unlikely that this incubation period would be insufficient for the expression of opsins at the terminals because it is standard to wait 3 weeks and because we can visually observe fluorescent expression in the BNST after 3 weeks when doing cell recordings.

3) The inclusion of the additional regions as negative findings, without any idea as if they were validated is problematic. I understand the author's position that this represents a great deal of work, but without further validation, these results are very difficult to appropriately interpret. This important caveat should be emphasized.

Response: This point has been emphasized in the Discussion.

4) The authors demonstrate on-off effects for Halorhodopsin animals are reversible across 6 test days, however they do not show this for control animals. The control animals should be plotted on the same graph as the Halorhodopsin animals is should be shown as it will allow readers to more clearly understand the null findings.

Response: The control animals are now plotted in the same graphs as the HaloR animals (Fig. 5, S5, S6, and S7).

5) The authors state "We also added no-virus controls for each CeA terminal output that was tested." However, in the figures they are listed as DIO-eYFP controls, please clarify. Performing light inhibition in the absence of a viral control is very unusual and the authors should provide a justification for doing so. Additionally, as these controls did not appear in the initial manuscript raises the question of whether they were run a later date. If so then this should be explicitly acknowledged in the manuscript. In addition, it is unclear if all of the other outputs were assessed at low drinking levels, as such this is an important caveat that should be included in the discussion.

Response: We agree that this was unclear. All of the controls in the manuscript were DIO-eYFP, and all of the DIO-eYFP animals were run in parallel with DIO-eNpHR animals. The only exceptions are the DIO-eYFP controls in Fig. 5, S5, S6, and S7, which were requested by the reviewers. Finally, we did not assess the effect of optogenetic inhibition of other CeA outputs in nondependent animals because we did not find any effect of optogenetic inhibition of the CeA in dependent animals. We now mention this caveat in the Discussion.

Minor concerns:

1. Figure 2A lacks any anatomical landmarks showing the extent of viral expression inside and outside of the CeA. In addition, the photomicrographs in figures 2, 4, 5, and the supplemental figures are all missing scale bars.

Response: The figures have been modified accordingly to the reviewer's suggestions.

References

Deng, Q., et al. (2015). "Rapid Glucocorticoid Feedback Inhibition of ACTH Secretion Involves Ligand-Dependent Membrane Association of Glucocorticoid Receptors." *Endocrinology* **156**(9): 3215-3227.

Kash, T. L., et al. (2008). "Dopamine enhances fast excitatory synaptic transmission in the extended amygdala by a CRF-R1-dependent process." J Neurosci **28**(51): 13856-13865.

Watanabe, T. and D. N. Orth (1987). "Detailed kinetic analysis of adrenocorticotropin secretion by dispersed rat anterior pituitary cells in a microperifusion system: effects of ovine corticotropin-releasing factor and arginine vasopressin." Endocrinology **121**(3): 1133-1145.

Reviewers' Comments:

Reviewer #3:

Remarks to the Author:

The revised manuscript is more clear and addresses my concerns.

We thank the reviewers for their very detailed and thoughtful comments on the original manuscript. As suggested, we performed a large number of additional experiments to address their concerns, including (1) testing behavioral controls using animals that were not injected with virus, (2) testing the effect of terminal inhibition using slice electrophysiology, (3) providing additional data analysis for the neuroanatomical characterization of Fos, (4) testing the effect of unilateral inhibition of the CeA on bilateral Fos counts, and (5) further characterizing the effect of the optogenetic inhibition of CRF neurons on action potentials. We believe the main conclusions of the revised manuscript (role of CRF^{CeA-BNST} neurons in alcohol drinking in dependent animals) were considerably strengthened by these additional experiments. We hope the revised manuscript addresses the reviewers' major concerns.

Reviewer 1

This manuscript presents experiments aimed at determining the role of neuronal ensembles in the central nucleus of the Amygdala (CeA) and associated downstream neural pathways in mediating aversive aspects of withdrawal from alcohol (EtOH), and in particular their role in escalated alcohol consumption as a form of presumed negative reinforcement. The authors use immunocytochemistry and optogenetics to identify the corticotropin releasing hormone (Crh) subset of CeA neurons and their connection specifically to the bed nucleus of the stria terminalis (BNST) as primary mediators of EtOH withdrawal symptoms and associated escalation of EtOH consumption. Specifically, the authors demonstrate that Crh containing CeA neurons make up the majority of activated CeA neurons (as assessed by immunostaining for the immediate early gene c-Fos) during EtOH withdrawal, and that selective optogenetic inhibition of such Crh neurons during EtOH withdrawal reduces CeA activation (both Crh and non-Crh neurons) and withdrawal symptoms, and, amazingly, completely eliminates escalated EtOH consumption. Subsequent selective optogenetic inhibition of Crh-positive CeA axon terminals in various efferent nuclei indicated that the BNST is the only output pathway to mediate such actions, and it does so to the same extent as CeA inhibition. All such impacts appear to be specific to EtOH withdrawal and withdrawal-induced escalated EtOH consumption, as evidenced by a variety of control experiments showing no impacts of optogenetic suppression of CeA and/or CeA to BNST afferents on basal EtOH consumption, water consumption, sucrose consumption, or locomotor activity. Based on these observations the authors conclude that the Crh containing neuronal ensemble within the CeA is the primary mediator of CeA activation, and via its connections to the BNST is the primary mediator of associated aversive withdrawal symptoms and consequent escalation of EtOH consumption. The topic of this manuscript is obviously of high social and clinical relevance, with alcohol abuse being a major burden on the health and socioeconomic fabric of our society. And, the social implications combined with the techniques used and circuitry mapped, should make this manuscript of interest to the wider readership. The rationale for the project, the approach and techniques are all very logically and clearly presented, and the specific experiments are, for the most part, well designed, executed, and presented. The authors have also emphasized implementation of a variety of important controls, in particular with respect to the use of optogenetics, which is important. Overall this is a very elegant study. There are, however, a few issues with the execution and presentation of several of the specific experiments, including control experiments (see specific comments for details). Briefly though, the methodology for quantifying the various immunocytochemistry studies is not very well described, and it is difficult, as presented, to be confident that the mean data accurately

reflects the raw data and presumed meaning of the raw data. Further, although the authors have emphasized their implementation of important control experiments for the optogenetic studies, which is commendable, there are some problems with the way they have been implemented, and key controls have not been done for the afferent studies, and that lack seriously compromises interpretation of the in vivo data. These issues should be addressed. Lastly, the results of this study are actually quite remarkable in how a single, arguably minor pathway, apparently completely mediates what is arguably a very complex scenario: EtOH withdrawal and associated escalated EtOH consumption. Consequently, further exploration and discussion of how this is possible is warranted (see specific comments for details).

Response: We thank the reviewer for the positive evaluation of the manuscript. We performed numerous additional experiments and revised the manuscript accordingly to address the concerns.

Issues:

1) Although the behavioral data for the studies of HaloR inhibition of CeA afferents to various nuclei is quite clear and striking (Fig. 5), important controls have not been done, and this seriously hampers overall interpretation of the data. Specifically, the authors need to test whether light activation of HaloR in specific afferent terminal targets actually inhibits axon terminals enough to significantly reduce synaptic transmission. This needs to be tested with slice studies, and ideally in vivo c-Fos studies. Such important controls need to be done in the proposed effective path (CeA to BNST), but especially in the noneffective pathways. It is difficult to inhibit action potential invasion of synaptic terminals, and hence vesicle release, with a small (5mV) hyperpolarization (and that assumes the hyperpolarization seen at the terminals is of the same magnitude at that observed at the soma, Fig. 2C&D?). This is important because although the behavioral data make a strong case for the importance of the CeA to BNST pathway, the conclusion that it is the only relevant pathway is completely dependent on assuming the light stimulation of HaloR in the other pathways is effective at inhibiting their target cell's excitation, which may not be the case, even if it is in the BNST afferents.

The authors should also provide some quantification of the estimated percent inhibition of the various nuclei tested (i.e. is the lack of effect in other nuclei simply due to the possibility that a smaller percent of total relevant neurons are inhibited). Such estimation could be based on volume of nuclei relative to estimated light volume, or better yet with actual quantification of percent silencing of c-Fos neurons in each nuclei.

Response: To address this important comment, we performed additional slice experiments. The results are presented in Fig. 5, showing that the inhibition of axon terminals in the BNST is sufficient to reduce neuronal activity, confirming the behavioral data. We revised the Discussion to read, "In slice experiments, we found that the inhibition of CeA CRF axon terminals in the BNST reduced the firing of a subpopulation of BNST neurons. This result is consistent with previous studies that reported that CRF can enhance spontaneous excitatory transmission in the BNST through an increase in glutamate release (Kash et al., 2008; Walker and Davis, 2008). A similar potentiation of excitatory transmission by CRF has been reported in the CeA and the lateral septum mediolateral nucleus (Liu et al., 2004)."

Although we were able to test the effect of the inhibition of axon terminals in the CRF^{CeA-BNST} projection using slice electrophysiology and confirm the validity of the optogenetic approach, we were unable to perform these experiments for all of the projections (i.e., CeA to LH, PBN, and SI). It took us ~9 months to perform the additional five experiments that are reported in the current revision of the manuscript. These remaining additional experiments are quite laborious and would require another 8-9 months of work (at best), and we would need to breed several new cohorts of animals (5 months) to obtain at least 30-40 animals, perform stereotaxic surgery (2 weeks), wait for the viral expression (1 month), and perform the electrophysiological recordings (2 months). Moreover, inhibition of the terminals through halorhodopsin has been successfully validated in previous studies (Seif, et al. 2013; Tye, et al. 2011).

Our laboratory has limited resources. We agree that these experiments would indeed be interesting, but we hope that the reviewers might agree that 18 months of revisions is an extremely long period of time, particularly because these experiments will not affect the main conclusion of the paper, which is that inhibiting the CRF^{CeA-BNST} projection decreases escalated alcohol drinking in dependent animals (note that we toned down the interpretation of the negative results and moved the negative results to the Supplementary Materials). Moreover, an important point is that we added 5 control experiments in this revised version of the manuscript, including an experiment for each brain region where we showed that light *per se* (in animals that were injected with control virus) did not affect alcohol intake in dependent animals. This is important control because it rules out possible nonspecific effects of optogenetic stimulation, further confirming our main findings on the CRF^{CeA-BNST} projection. Finally, as mentioned by one of the reviewers, slice electrophysiology does not necessarily reflect *in vivo* conditions. Even if we performed these additional experiments and observed positive results, one could argue that it does not reflect physiological conditions. We again hope that the reviewers will agree that adding 9 months of additional work is unnecessary to confirm the negative results (which were not the main focus of the study). We believe that these new data with the control virus, together with the electrophysiological data that show a reduction of neuronal activity in the BNST after the light inhibition of CRF^{CeA-BNST} terminals, represent a proper validation of our model and solid controls for our main findings.

2) Related to the comment above, the most striking result of this study is that light activated HaloR in the CeA to BNST terminals (unilaterally) is enough to completely inhibit withdrawal-induced escalated EtOH consumption (Fig. 5B), despite having no impact on three of the 5 withdrawal symptoms tested (Fig. 5C; and unknown impact on other withdrawal symptoms). This raises the question of how such inhibition is causing such an outcome? The way the manuscript is introduced, one might infer that the mechanism is envisioned to be reduced withdrawal symptoms, and thus less drive for EtOH consumption as a negative reinforcer. However, this does not seem to fit with the fact that many withdrawal symptoms that are presumably aversive persist (Fig. 5C). An alternative possibility that is not considered is that the circuit plays a role in associative learning, in a manner that inhibition of the circuit inhibits memory of the ameliorative aspects of EtOH, which might then be expected to eliminate escalated EtOH consumption despite persistence of various aversive symptoms. It would increase the impact of this manuscript a great deal if the authors could discriminate between these two possibilities. Another possibility is that EtOH itself only reduces some of the aversive withdrawal symptoms, most parsimoniously the ones that are eliminated when CeA to BNST afferents are inhibited (abnormal gait & body tremors), and so optogenetic inhibition effectively

provides the same relief as EtOH, both being incomplete but equally reinforcing. It would be informative to determine what EtOH does to the tested withdrawal symptoms.

Response: These are interesting points. We incorporated them into the revised Discussion to read, “The present results also showed that the inhibition of CeA terminals in the BNST completely inhibited the withdrawal-induced escalation of alcohol consumption in dependent rats. The mechanism for this marked reduction of alcohol drinking might be related to a decrease in withdrawal symptoms that leads to less negative reinforcement associated with alcohol consumption. However, inhibition had no impact on three of the five withdrawal signs that were evaluated. One explanation for this result may be that somatic signs provide behavioral evidence of alcohol withdrawal, but they do not directly measure the negative affective state that is thought to drive the escalation of drinking. Another possibility may be that the CRF^{CeA-BNST} pathway plays a role in associative learning, and the inhibition of this pathway inhibits memories of the ameliorative effects of alcohol. Such a mechanism could be involved in mitigating the escalation of alcohol consumption despite the persistence of some withdrawal signs. Further studies are needed to discriminate between these different possibilities.”

3) A final concern, related to comments 1&2 above, is the fact that unilateral inhibition of the Crh CeA neurons or their afferents to the BNST completely prevents escalated EtOH consumption (Fig. 4B). This is somewhat surprising. Specifically, it is difficult to understand this complete block of escalated consumption in the context of the overall proposed model, i.e. that CIE-induced altered Crh-CeA signaling underlies the aversive aspects of withdrawal that drive escalated EtOH consumption as a negative reinforcer. Do the authors propose that one hemisphere of such alterations has no aversive impact? Perhaps relatedly, in Fig. 4H it appears that the image of the CeA is a mirror image orientation of the images in Fig4F&G (based on the position of the CeA relative to the direction of the tip of the nearby fissure). Is this the contralateral hemisphere? Either way it would be informative to examine C-Fos expression in the contralateral amygdala. Does unilateral inhibition somehow inhibit both hemispheres? If not, the authors should at least discuss this issue.

Response: We addressed this important point by providing an analysis of Fos expression in the contralateral CeA during optogenetic inhibition. The results showed that unilateral inhibition also affected the contralateral CeA. We observed a significant decrease in Fos⁺ neurons on the contralateral side. These results suggest that the reviewer is correct and that the CeA may be functionally connected bilaterally so that if one side is inhibited, then the other side is also inhibited. These data are now presented in Fig. 4.

4) The authors quantify EtOH consumption by “number of rewards” (presumably the number of active lever presses) during Operant self-administration trials. While this is an understandably easily quantified measurement, it does not inform us of the blood EtOH concentrations (BECs) achieved during such trials, which has several important ramifications and/or drawbacks. First, it does not seem appropriate to refer to such measurements as “excessive drinking”, page 4, 3rd sentence. All we can say is that drinking increased. Second, and more importantly, in the context of the EtOH consumption and escalation during EtOH withdrawal, it is implied that the driving force for elevated consumption is amelioration of the associated aversive state. Thus it is important to determine whether BECs achieved are actually high enough to significantly alter

CeA signaling. Such information might partially inform issue #2 above. Additionally, it is important to know what potential molecular mediators of such action are, which requires knowing the BECs achieved. I.e. are the levels achieved likely able to suppress CeA NMDARs, enhance GABAARs, alter transmitter release, alter Crh, etc. Thus, for both aspects, it is important to determine and present BECs achieved during baseline and elevated EtOH consumption.

Response: We now express the drinking data as g/kg of alcohol. Our dependent animals drank >1 g/kg per 30 min session. These amounts of alcohol have been shown to produce BALs of ~0.2 g%, which is known to produce multiple adaptations to the stress regulatory system, compromised hormonal responses, and sensitized brain stress responses in dependent animals (Richardson, et al. 2008).

5) There are a variety of issues that make it difficult to assess the outcomes of the immunocytochemical analysis and the relationship between the raw data and mean data as presented in the bar graphs (Figs 1E&F and Suppl. Fig. 1 C&D, and Fig. 2B). In particular, there is a considerable amount of ambiguity about the nature and magnitude of the staining to the naked eye in the raw data images, and there is an inadequate description of the quantitative details of how bar graphs were generated from such ambiguous images. There is also a lack of controls showing staining is specific and/or represents what the authors express in the bar graphs. Of specific concern in Fig. 1E&F is how the authors determined what counted as a c-Fos positive or negative cell. First, proper immunocytochemical controls showing that staining is specific to c-Fos are needed, or at least a citation to prior work showing that the specific antibody used is specific. Second, while the arrows appear to point to what the authors conclude are c-Fos positive cells (note, although deducible, the legend lacks a description of what the arrows and their color signify), there is clearly abundant green cytoplasmic staining in a much larger proportion of the exhibited cells but with a different apparent morphology (apparent cytoplasmic rings versus solid circles of color). What are the respective staining patterns? Is this an artifact of confocality (with rings versus solid circles depending on depth into a given cell)? Is the antibody specific? If the antibody is specific, what are the apparent cytoplasmic rings, and do they change (from the exemplar images it looks like rings disappear in withdrawal animals, does this reflect redistribution of c-Fos?), and what does this mean? Similarly, in Supplemental Fig. 1 C&D, while one can get a vague sense that arrows point to “darker” spots, it is not by any means clear. Basically every nucleus has some degree of brownish/black, and there is no description of how thresholding was determined or implemented, or how the accuracy of such implementation was confirmed? Furthermore, the dark spots do not appear to co-localize with the blue stain (presumed NeuN, indicative of neuronal subtype), which is supposed to indicate neuronal overlap. Again, a better description of methodology and confirmation of validity is needed.

Response: We thank the reviewer for these detailed comments. We clarified the immunohistochemistry methods by providing additional details on the validity of the antibody and analyses. We also provided additional images to address these concerns.

(1) The antibody that was used to detect Fos (Cell Signaling Technology, catalog no. 2250) has been used in over 71 publications as a specific marker for Fos (<https://www.citeab.com/antibodies/123097-2250-c-fos-9f6-rabbit-mab/publications>), as well as in our previous publications (de Guglielmo, et al. 2016; Leao, et al. 2015).

(2) We agree that there was a lot of background in the previous images. We now provide better representative photographs that are clearer. We classified neurons as Fos+/CRF+ when a neuron exhibited a focused red nucleus (Fos+) and a cytoplasm that was stained for CRF (green). Yellow arrows indicate Fos-/CRF+ cells. White arrows indicate Fos+/CRF+. Orange arrows indicate Fos+/CRF-. Three sections from each rat were bilaterally quantified within the boundary of the CeA and normalized by area, measured using Fiji software. Counts from all images for each rat were averaged so that each rat was an n of 1. The immunohistochemistry section in the Methods was revised to include this information.

(3) For DAB staining, brown (NeuN) and black (Fos) channels were separated using Fiji software to isolate the red channel. Using the red channel as the background channel for thresholding leaves the dark nuclei from the black (Fos) channel easy to identify and threshold. Thresholding was applied under the same conditions for both naive and alcohol-withdrawn rats. Thresholded images were counted for positive foci within the nucleus of NeuN-positive neurons. The immunohistochemistry section in the Methods was revised to include this information, and we added a sample image to Supplementary Fig. S1 (Fig. S1C).

6) There are some important issues with the electrophysiological analysis of the impact of HaloR activation on CeA neuronal excitability. First, the language describing the studies is confusing. The recordings are clearly in the current clamp configuration (i.e. clamping electrode current to record membrane potential), but the authors refer to holding potential (last sentences on page 5) which implies voltage-clamp (i.e. clamping membrane potential to record currents). This language needs to be fixed.

Response: We apologize for this mistake. The reviewer is correct that the experiment was performed in current-clamp mode. We rephrased this section of the Results and removed “held at” or “holding potential” to avoid confusion.

More importantly, there is no description of what the resting membrane potential was (and whether it was similar across conditions), or how much current was injected to alter the membrane potential to what one presumes was designed to be similar across cells?

Response: We now indicate the average resting membrane potential for the two groups. There was no significant difference between the two groups. We also added the average input resistance. There was no significant difference between the two groups; hence, similar current injections were needed to obtain similar depolarizations.

Most importantly though, this being a crucial control experiment, as best I can tell the authors have injected current to adjust the membrane potential to be right above action potential (AP) threshold, hence the steady action potential firing. While this approach provides some information (i.e. it clearly shows that light excitation of HaloR slightly hyperpolarizes CeA cells, and if they are just above AP threshold, it can silence their AP firing), it does not tell us much about the likely impact of light-activation of HaloR under normal physiological conditions, when cells are not manipulated, by constant current injection, to be just above AP threshold. What is more relevant is what impact such a small hyperpolarization has on post synaptic AP responses to signals from other cells, either in the form of glutamatergic/GABAergic or CRF transmission. This is particularly important given the very small hyperpolarization (5mV) induced by light

activation of HaloR. The authors should determine whether light excitation of HaloR can reduce AP output induced by afferent stimulation or CRF application, when the cells are not artificially depolarized with current injection.

A simpler but less elegant alternative, would be to examine the impact of light inhibition on responses to transient current injections from a cell at its native resting potential.

Response: This experiment was designed to confirm the validity of the optogenetic approach. As the reviewer pointed out, while having some drawbacks, the approach “clearly shows that light excitation of HaloR slightly hyperpolarizes CeA cells.” The spontaneous firing of CeA neurons at rest varies from cell to cell. Therefore, it is important to normalize these variations to be able to interpret the results without having to record an unreasonable number of cells. With intracellular recordings, spontaneous firing at rest can vary considerably from cell to cell. By depolarizing neurons, we ensured consistent and sustained firing to observe the influence of NpHR on firing rate. Additionally, we believe that a hyperpolarization of 6-7 mV qualifies as a strong effect rather than “very small,” although it is more likely to be 4-5 mV around the resting potential (still a sizeable effect). Being able to directly correlate the degree of hyperpolarization and decrease in spiking with behavioral observations would be exciting, but we feel that it would be quite a stretch. It must be kept in mind that we are using an *ex vivo* approach. Although brain slices can last several hours in excellent health, we remain far from “normal physiological conditions.” Consequently, one could argue that the resting potential of a neuron that is seen in the slice preparation may be different from the resting potential that is seen *in vivo*, and a non-spiking neuron in a slice could be a spiking neuron *in vivo*. Thus, pinpointing the effect of NpHR at resting potential to build an interpretation for the behavioral observations carries significant caveats. We agree that the alternative approaches that are proposed by the reviewer could provide additional information, but they would not change the original results and demonstration of the validity of the optogenetic approach in silencing CRF+ neurons that are firing.

Also, in both the existing experiment and any future experiments, the authors should provide detailed quantification of the change in AP output, as opposed to just stating that it “suppressed AP firing rate” (line 95).

Response: We now quantify the changes in action potential output.

I emphasize, combined with the behavioral data I don't doubt that the HaloR stimulation can reduce CeA firing *in vivo*, but it is important to know more realistically how such behavioral changes are induced. As the figure stands it might be concluded that HaloR stimulation completely shuts down CeA firing, which is highly unlikely to be the case under more physiological circumstances. This is important for our understanding of how much cellular inhibition is required to implement such dramatic behavioral outcomes.

Response: We fully agree with the reviewer that the all-or-none figure may lead the reader to conclude that HaloR shuts down CeA activity. We quantified the reduction of action potential firing by HaloR and now emphasize “reduced spiking activity” with HaloR. Additionally, our new results in the BNST show lower spiking activity instead of a shutdown (see Fig. 5).

7) The graphical display of withdrawal symptom subsets is too small (Figs. 4C and 5C-L). While I appreciate the value of the combined score insert, the potentially more informative comparisons of withdrawal subset comparisons across Crf-CeA inhibition and CeA to BNST afferent inhibition is very difficult to see given the scaling required to accommodate the insert. The Y-axis scale of the subset graphs should be expanded so that comparisons across the two conditions can be more easily made. The large size of combined totals insert, while informative, obscures differences in the subcategories. In this regard, it appears that the only subcategory that is significantly reduced by both optical inhibition of the Crf-CeA and CeA to BNST afferents is “abnormal gait”, despite both completely inhibiting escalated EtOH consumption. Does this imply abnormal gait is the main behavioral manifestation of EtOH withdrawal that drives escalation of EtOH consumption? If so, how does this relate to the known functions of CeA to BNST circuit, and/or downstream aspects of the circuit?

Response: As suggested by the reviewer, the Y-axis scale for the individual signs was expanded. However, the withdrawal signs are very variable between individuals, and this is the reason why we and other groups show the sum of the individual signs as a “Total Withdrawal Score.” For references, see (Braconi, et al. 2010; de Guglielmo, et al. 2016; de Guglielmo, et al. 2017b; de Guglielmo, et al. 2015; Macey, et al. 1996; Sidhpura, et al. 2010). It is important to clarify that we do not believe that there is necessarily a causal relationship between the reductions of some of the somatic signs and alcohol drinking. We clarified this point in the Discussion.

Minor comments:

1) The authors need to refer to a manuscript that describes the generation and characterization of the Crh-Cre rats used in this study. Apologies if I missed it somewhere, but it is not in the first reference to the animals or in the methods that describe their use as subjects.

Response: We cited the following article in the Introduction and Electrophysiology and Subjects sections of the Methods: Pomrenze (2015) A Transgenic Rat for Investigating the Anatomy and Function of Corticotrophin Releasing Factor Circuits. *Front Neurosci* 9, 487.

2) The authors should explicitly state the precise time frame (relative to CIE termination) for when operant self-administration and withdrawal symptoms were quantified.

Response: This information is in the Methods section (Alcohol vapor chambers):
“...the rats were made dependent by chronic intermittent exposure to alcohol vapors. They underwent cycles of 14 h ON (blood alcohol levels during vapor exposure ranged between 150 and 250 mg%) and 10 h OFF, during which behavioral testing for acute withdrawal occurred (i.e., 6-8 h after the vapor was turned OFF, when brain and blood alcohol levels are negligible).”

Reviewer 2

The manuscript by George and colleagues seeks to assess the role for neuronal activations of the central nucleus of the amygdala (CeA) aspects of alcohol behaviors that are associated with dependence/compulsive drinking in rodent models. The overall hypothesis was that corticotropin-releasing factor (CRF) neurons and downstream targets, such as their known

projections to the bed nucleus of the stria terminalis (BNST), might be involved and that these CRF neurons and related circuitry might be a potential target for reducing compulsive drinking in AUD. These authors have utilized a number of state-of-the-art techniques to test their hypothesis, and have provided a well written and compelling study. The hypothesis that the CeA is involved in AUD is solidly based on a large pre-clinical literature, notably that neuronal ensembles in the CeA are associated with alcohol withdrawal behaviors in alcohol-dependent rats. A large body of evidence implicates CRF neurotransmission in the CRF in withdrawal and negative emotional states, and supportive pharmacological evidence for a CRF CeA link. In this study, first Crh-Cre transgenic rats were used in an animal model of alcohol dependence combined with in vivo optogenetics and immediate early gene brain mapping. Then the role for several possible downstream CeA-CRF projections were assessed by optogenetic inactivation methods, i.e., CRF terminals from CeA neurons that project to the BNST (CRFCeA-BNST), LH and pSTN (CRFCeA-LH/pSTN), substantia innominata (SI; CRFCeA-SI), and parabrachial nucleus (PBN; 59 CRFCeA-PBN).

The main findings of these experiments were that 1) withdrawal from alcohol dependence significantly and mostly affects CRF neurons (~ 80%), 2) that the inactivation of these CeA CRF+ neurons prevents the recruitment of the CeA neuronal ensemble, and reduces dependence-associated alcohol drinking and withdrawal, and 3) that based on the optogenetic circuitry assessment that it is activation of CeA neurons that project to the BNST that are critically involved. The paper is exceptionally well prepared and generally presented.

Response: We thank the reviewer for the positive evaluation of the manuscript.

There are several issues of interpretation and some details that need to be addressed. The conclusion that CRF neurons are critical to “drive compulsive-like alcohol drinking” is unclear. Although dependent rats were found to have a significant proportion of the CeA neuronal ensemble that is associated with withdrawal in alcohol-dependent rats, it seems a stretch to argue that this reflects compulsive (or compulsive-like) behavior, or what component of “compulsion”. While these data do seem to support the overall conclusion that there are key CeA neuronal ensembles that are involved in alcohol drinking in dependent rats – nor is clear if this is only a result from the vapor-inhalation model. These data do nonetheless show that inactivation of the CRFCeA-BNST pathway – rather selectively -- could reverse this “excessive” alcohol drinking during withdrawal and partially prevented the somatic signs of withdrawal. More discussion and validation is needed to argue that these data have identified a cellular mechanism of excessive alcohol use, and it appears that with regards to this statement in the Introduction they cite their own JN paper. Indeed, in the Results the word “escalation” of intake is used to describe these actual data (Figure 1A), though then at other points it is then referred to as “excessive”, and then as noted above, the term “compulsive” then gets introduced when these -- though the title and one place in the Discussion uses the careful “addiction-like” term, even though the Discussion starts with “compulsive-like”. I think that words matter.

Response: We thank the reviewer for this comment. We now refer only to the escalation of alcohol intake.

The details with regards to numbers of animals used for the statistical analyses are not transparent in several places. In fact (perhaps I missed it) I cannot find any details with regards to

number of animals for each experiment/condition with the exception of the whole-cell recordings. Statistical analyses section is too general and does not make it clear the numbers of animals that were used/reported in these data presented in the figure legends.

Response: The number of rats used in each experiment is now stated in the text.

A number of minor issues need to be addressed.

It would be helpful that scale the axis for comparisons to be matched. For example in Figure 5 the axis showing the rewards with the optogenetic inhibition of the inhibition BNST is not the same as for the SI, LH, and PBN for the alcohol SA data.

Response: As suggested by Reviewer 3, we reported all of the alcohol data as grams of alcohol per kilogram of body weight. We also ensured that all scales are the same for axis comparisons.

There is no justification for the exclusive use on male rats, and given NIH guidelines female rats should be included.

Response: The integration of female data would be really interesting, but it is beyond the scope of the present study. This project started several years before NIH's new guidelines on female studies, and it would take several years to perform all of these experiments again in females. We added a sentence in the manuscript to address this limitation.

There are scant details with regards to the operant self-administration and methods used to “get” animals to drink the 10% (w/v) alcohol, and for the saccharin self-administration study, and whether the alcohol animals were initially trained to drink saccharin and alcohol in a “fading” design.

Response: The animals were not pretrained to lever press for saccharin and did not undergo a fading procedure. The self-administration training is very straightforward and is described in the methods “For the alcohol self-administration studies, the animals were first trained to self-administer 10% (v/v) alcohol and water solutions until a stable response pattern (20 ± 5 rewards) was maintained. The rats were subjected to an overnight session in the operant chambers with access to one lever (right lever) that delivered water (fixed-ratio 1 [FR1]). Food was available *ad libitum* during this training. After 1 day off, the rats were subjected to a 2 h session (FR1) for 1 day and a 1 h session (FR1) the next day, with one lever delivering alcohol (right lever). All of the subsequent alcohol self-administration sessions lasted 30 min. The rats were allowed to self-administer a 10% (v/v) alcohol solution (right lever) and water (left lever) on an FR1 schedule of reinforcement (i.e., each operant response was reinforced with 0.1 ml of the solution).”

These studies are based on a model of alcohol dependence produces by exposure to chronic intermittent alcohol vapor. While many groups use this model, it is not clear to the extent this model has clear translational relevance and that these effects can be ascribed selectively to alcohols pharmacological effects as opposed to more general effects of stress or aversion. For example, while the number of Fos+/CRF+ neurons in the CeA significantly increased during withdrawal this is based on the comparison between alcohol-withdrawn and alcohol-naive rats

and in my opinion the alcohol-naïve rats are not an appropriate control. Given the above concern, the authors could use other alcohol drinking models other than vapor inhalation.

Response: The model of chronic intermittent exposure (CIE) to alcohol vapor has been shown to have robust predictive validity for alcoholism and construct validity for the neurobiological mechanisms of alcohol dependence (Heilig and Koob 2007; Koob 2009). Indeed, rats that are made dependent by CIE exhibit clinically relevant blood alcohol levels (BALs; 150-250 mg/100 ml), an increase in alcohol drinking when tested during early and protracted abstinence, and importantly compulsive-like alcohol drinking (e.g., responding despite adverse consequences; (Kimbrough, et al. 2017b; Leao, et al. 2015; O'Dell, et al. 2004; Roberts, et al. 1996; Schulteis, et al. 1995; Vendruscolo, et al. 2012). Alcohol dependence that is induced by alcohol vapor also results in withdrawal symptoms during both acute withdrawal and protracted abstinence (de Guglielmo, et al. 2017a; Kallupi, et al. 2014; Macey, et al. 1996; Vendruscolo and Roberts 2014), anxiety-like behavior (Valdez, et al. 2002), irritability-like behavior (Kimbrough, et al. 2017a), and the development of mechanical hyperalgesia (Edwards, et al. 2012). With regard to the increase in Fos+/CRF+ neurons during withdrawal in dependent rats, we understand that the reviewer feels that a nondependent control would have been more appropriate. However, it has been shown that withdrawal from alcohol in nondependent subjects does not induce Fos activation in the CeA (George, et al. 2012; Weitemier, et al. 2001). In the present study (Fig. 3A), we found that the inactivation of CeA CRF neurons did not affect alcohol intake in nondependent rats.

These whole-cell patch electrophysiological data to confirm the efficacy of NpHR in preventing CRF+ neuron firing in acute CeA slices from Crh-Cre rats was not well described or integrated into the manuscript.

Response: The electrophysiology section has been reorganized and improved to provide additional details and data (see also response to Reviewer 1).

It is confusing that the Figures with legends are presented before the References cited, and then shown again without the legends at the end of the manuscript.

Response: The manuscript has been reorganized, and the figure legends are presented only once.

Reviewer 3

Major Concerns:

1) The authors examine many potential outputs of CeA CRF neurons including the Bed Nucleus of the Stria Terminalis (BNST), Substantia Inominata (SI), Lateral Hypothalamus (LH), and Parabrachial Nucleus (PBN). While some of these outputs are well-known, such as the BNST and PBN, others, such as the SI and LH, have not been described previously. Given light microscopy cannot differentiate between terminals and fibers of passage, it is unclear whether the lack of effects seen in these regions are true effects or are simply without effect due to the lack of synapses present. The authors must test the function of these outputs in order to claim that inhibition of these outputs are without effect. One potential way to resolve this question

would be to perform light-evoked recordings in this region and examine what percentage of cells are responsive to stimulation.

2) Going further, what is the overlap between BNST-projecting CRF neurons and the other outputs tested? Are BNST projections are separate population or are they collaterals of the same cells?

Response: These are very interesting points. We agree that the negative results with the other pathways could potentially be attributable to either fiber of passage or technical limitations, or these pathways may play roles in other withdrawal-related behaviors. However, testing these novel hypotheses would require several years of work and are beyond the scope of the present study. Moreover, whether the BNST projections are separate populations or collaterals of the same cells would not affect the main conclusion of the study, which is that CeA-to-BNST CRF neurons are required for excessive alcohol drinking and withdrawal symptoms in dependent rats. We toned down the negative results to emphasize the positive results that were obtained with the BNST, for which we performed an additional experiment to test the effect of inhibition of the terminals. We moved the negative results from the main section of the manuscript to the Supplementary Material to ensure transparency and because we believe that this information may still be very useful to other laboratories that work on these pathways.

3) The authors find that both somatic inhibition and inhibition of BNST outputs produce similar effects. This does not preclude a role for local CRF neurons in the CeA in regulating alcohol intake. Given the body of work describing the effects of alcohol on CRFR1+ cells in the CeA (Herman et al. 2013; Herman et al. 2016), it would be helpful to examine a role for CRF neurons in the CeA using an intersectional strategy to target non-BNST projecting neurons. A major limitation of this paper is lack of any evidence describing a role for CRF signaling itself. The authors acknowledge that CeA CRF neurons also express a medley of other neuropeptides such as dynorphin, neotensin, and somatostatin, all of which may be contributing to the effects seen. Thus it is unclear the whether the results seen are due to manipulation of CRF signaling or inhibition of a unique subset of CeA neurons.

Response: The hypothesis of the study was not to demonstrate that CRF peptide is the sole contributor to alcohol drinking. We fully agree with the reviewer that other neurotransmitters and neuromodulators very likely contribute to the observed effects. We tested the role of CRF expressing neurons and their pathways in alcohol drinking. Dissecting the role of the different peptides and neurotransmitters within these neurons is a very interesting but monumental task and beyond the scope of the present study.

Another major limitation of this study is the lack of appropriate non-dependent controls. Studies using CIE should include air exposed controls to show that effects are due induction of alcohol-dependence. More seriously, there are no virus controls for the experiments in figure 5. Without these data points, it is not possible to determine whether the experiments in this aim are the result are a true effect of inhibition of the pathway or a spurious result of laser stimulation.

Response: The effect of inhibiting CeA CRF neurons in nondependent subjects is described in Fig. 3A. We showed that halorhodopsin inhibition of CRF neurons in the CeA did not affect alcohol self-administration in nondependent rats. We also added no-virus controls for each CeA

terminal output that was tested. We found that light activation of each of the projection sites did not affect alcohol self-administration in dependent rats.

The author claim they targeted CRF neurons in the CeL, however there is also a population of CRF neurons in the CeM. Given the lack of clear anatomical boundaries between these two subdivisions, it is difficult to believe that expression was constrained to the CeL.

Response: We agree with the reviewer and now use the term CeA throughout the paper instead of CeL.

Throughout the manuscript statistics are poorly described, and in several cases do not match the data shown.

Response: We carefully checked and updated all of the statistics and improved the descriptions of the results. The statistical analysis paragraph in the methods section now reads: “The data are expressed as mean \pm SEM. For comparisons between only two groups, p values were calculated using paired or unpaired t -tests as described in the figure legends. Comparisons across more than two groups were made using one-way analysis of variance (ANOVA), and two-way ANOVA was used when there was more than one independent variable. Experiments that tested the effect of the unilateral optogenetic inhibition of CeA CRF neurons on the escalation of alcohol self-administration in dependent rats were analyzed using a two-way repeated-measures ANOVA, with time and laser illumination as within-subjects factors (see Fig. 4, 5, S4, S5, and S6). In the same experiments, differences between baseline intake before dependence and intake after vapor exposure (ON and OFF days; see Fig. 4, 5, S4, S5, and S6) were analyzed using separate one-way repeated-measures ANOVAs. The Newman Keuls *post hoc* test was used following significance in the ANOVA. Withdrawal signs were analyzed using the nonparametric Mann-Whitney U statistic, followed by Dunn’s multiple-comparison test. The standard error of the mean is indicated by error bars for each group of data. Differences were considered significant at $p < 0.05$. All of the data were analyzed using Statistica 7 software.

The authors do not demonstrate effective presynaptic inhibition using the in vivo parameters. This is a topic of great debate and needs to be included.

Response: This is a very important point, and we performed slice experiments to address this issue. The results are presented in Fig. 5, showing that the inhibition of axon terminals in the BNST is sufficient to reduce neuronal activity, confirming the behavioral data.

Minor Concerns:

For introduction cite original research papers rather than reviews?

Response: Citations from original research papers have been added to the Introduction.

What quantity of alcohol is administered by the rats? All the figures show the number of rewards obtained, however it would be more informative to report this as grams of ethanol per kilogram of body weight. This allows the readers to understand the effective dose of alcohol animals are receiving and would normalize differences in body weight across animals.

Response: All of the alcohol data are now expressed as grams of alcohol per kilogram of body weight, as suggested by the reviewer.

The Whole-Cell Patch Clamp recording section begins rather abruptly. The manuscript would read more smoothly if a few sentences of introduction were added stating that the purpose of these experiments were to validate the efficacy of DIO-NpHR in the CeA of CRF-Cre rats. Likewise, it would be helpful to have a more descriptive title such as “Halorhodopsin effectively inhibits CeA CRF neurons” rather than just calling it “Whole-Cell Patch Clamp Recording”

Response: The electrophysiological validation of the efficacy of NpHR in preventing CRF+ neuron firing section has been modified accordingly to the reviewer’s suggestion.

In the methods section, a specific age range of animals should be given rather than listing them as “adults”

Response: Done.

References

- Braconi, S., et al.
2010 Revisiting intragastric ethanol intubation as a dependence induction method for studies of ethanol reward and motivation in rats. *Alcohol Clin Exp Res* 34(3):538-44.
- de Guglielmo, G., et al.
2016 Recruitment of a Neuronal Ensemble in the Central Nucleus of the Amygdala Is Required for Alcohol Dependence. *J Neurosci* 36(36):9446-53.
- de Guglielmo, G., et al.
2017a Voluntary induction and maintenance of alcohol dependence in rats using alcohol vapor self-administration. *Psychopharmacology (Berl)* 234(13):2009-2018.
- 2017b Voluntary induction and maintenance of alcohol dependence in rats using alcohol vapor self-administration. *Psychopharmacology (Berl)*.
- de Guglielmo, G., et al.
2015 MT-7716, a potent NOP receptor agonist, preferentially reduces ethanol seeking and reinforcement in post-dependent rats. *Addict Biol* 20(4):643-51.
- Edwards, S., et al.
2012 Development of mechanical hypersensitivity in rats during heroin and ethanol dependence: alleviation by CRF(1) receptor antagonism. *Neuropharmacology* 62(2):1142-51.
- George, O., et al.
2012 Recruitment of medial prefrontal cortex neurons during alcohol withdrawal predicts cognitive impairment and excessive alcohol drinking. *Proc Natl Acad Sci U S A* 109(44):18156-61.
- Heilig, M., and G. F. Koob
2007 A key role for corticotropin-releasing factor in alcohol dependence. *Trends Neurosci* 30(8):399-406.

- Kallupi, M., et al.
2014 Neuropeptide YY(2)R blockade in the central amygdala reduces anxiety-like behavior but not alcohol drinking in alcohol-dependent rats. *Addict Biol* 19(5):755-7.
- Kimbrough, A., et al.
2017a CRF1 Receptor-Dependent Increases in Irritability-Like Behavior During Abstinence from Chronic Intermittent Ethanol Vapor Exposure. *Alcohol Clin Exp Res*.
- Kimbrough, A., et al.
2017b Intermittent Access to Ethanol Drinking Facilitates the Transition to Excessive Drinking After Chronic Intermittent Ethanol Vapor Exposure. *Alcohol Clin Exp Res* 41(8):1502-1509.
- Koob, G. F.
2009 Brain stress systems in the amygdala and addiction. *Brain Res* 1293:61-75.
- Leao, R. M., et al.
2015 Chronic nicotine activates stress/reward-related brain regions and facilitates the transition to compulsive alcohol drinking. *J Neurosci* 35(15):6241-53.
- Macey, D. J., et al.
1996 Time-dependent quantifiable withdrawal from ethanol in the rat: effect of method of dependence induction. *Alcohol* 13(2):163-70.
- O'Dell, L. E., et al.
2004 Enhanced alcohol self-administration after intermittent versus continuous alcohol vapor exposure. *Alcohol Clin Exp Res* 28(11):1676-82.
- Richardson, H. N., et al.
2008 Alcohol self-administration acutely stimulates the hypothalamic-pituitary-adrenal axis, but alcohol dependence leads to a dampened neuroendocrine state. *Eur J Neurosci* 28(8):1641-53.
- Roberts, A. J., M. Cole, and G. F. Koob
1996 Intra-amygdala muscimol decreases operant ethanol self-administration in dependent rats. *Alcohol Clin Exp Res* 20(7):1289-98.
- Schulteis, G., et al.
1995 Decreased brain reward produced by ethanol withdrawal. *Proc Natl Acad Sci U S A* 92(13):5880-4.
- Seif, T., et al.
2013 Cortical activation of accumbens hyperpolarization-active NMDARs mediates aversion-resistant alcohol intake. *Nat Neurosci* 16(8):1094-100.
- Sidhpura, N., F. Weiss, and R. Martin-Fardon
2010 Effects of the mGlu2/3 agonist LY379268 and the mGlu5 antagonist MTEP on ethanol seeking and reinforcement are differentially altered in rats with a history of ethanol dependence. *Biol Psychiatry* 67(9):804-11.
- Tye, K. M., et al.
2011 Amygdala circuitry mediating reversible and bidirectional control of anxiety. *Nature* 471(7338):358-62.
- Valdez, G. R., et al.
2002 Increased ethanol self-administration and anxiety-like behavior during acute ethanol withdrawal and protracted abstinence: regulation by corticotropin-releasing factor. *Alcohol Clin Exp Res* 26(10):1494-501.
- Vendruscolo, L. F., et al.

- 2012 Corticosteroid-dependent plasticity mediates compulsive alcohol drinking in rats.
J Neurosci 32(22):7563-71.
- Vendruscolo, L. F., and A. J. Roberts
2014 Operant alcohol self-administration in dependent rats: focus on the vapor model.
Alcohol 48(3):277-86.
- Weitemier, A. Z., et al.
2001 Expression of c-Fos in Alko alcohol rats responding for ethanol in an operant
paradigm. Alcohol Clin Exp Res 25(5):704-10.